# Neurogenomic insights into paternal care and its relation to territorial aggression

Syed Abbas Bukhari[1,2,3], Michael C. Saul[1,6], Noelle James [4], Miles K. Bensky[5], Laura R. Stein[3,7], Rebecca Trapp[3,8] & Alison M. Bell[1,3,4,5]

Motherhood is characterized by dramatic changes in brain and behavior, but less is known about fatherhood. Here we report that male sticklebacks—a small fish in which fathers provide care—experience dramatic changes in neurogenomic state as they become fathers. Some genes are unique to different stages of paternal care, some genes are shared across stages, and some genes are added to the previously acquired neurogenomic state. Comparative genomic analysis suggests that some of these neurogenomic dynamics resemble changes associated with pregnancy and reproduction in mammalian mothers. Moreover, gene regulatory analysis identifies transcription factors that are regulated in opposite directions in response to a territorial challenge versus during paternal care. Altogether these results show that some of the molecular mechanisms of parental care might be deeply conserved and might not be sex-specific, and suggest that tradeoffs between opposing social behaviors are managed at the gene regulatory level.

[1] Carl R. Woese Institute for Genomic Biology, University of Illinois, Urbana Champaign, 1206 Gregory Drive, Urbana, IL 61801, USA. [2] Illinois Informatics Institute, University of Illinois, Urbana Champaign, 616 E. Green St., Urbana, IL 61820, USA. [3] Department of Evolution, Ecology and Behavior, University of Illinois, Urbana Champaign, 505 S. Goodwin Avenue, Urbana, IL 61801, USA. [4] Neuroscience Program, University of Illinois, Urbana Champaign, 505 S. Goodwin Avenue, Urbana, IL 61801, USA. [5] Program in Ecology, Evolution and Conservation Biology, University of Illinois, Urbana Champaign, 505 S. Goodwin Avenue, Urbana, IL 61801, USA. [6] Present address: Jackson Labs, 600 Main St., Bar Harbor, ME 04609, USA. [7] Present address: Department of Biology, University of Oklahoma, 730 Van Vleet Oval, Room 314, Norman, OK 73019, USA. [8] Present address: Department of Biological Sciences, Purdue University, 915 W. State St., West Lafayette, IN 47907, USA. Correspondence and requests for materials should be addressed to A.M.B. (email: alisonmb@illinois.edu)

In many species, parents provide care for their offspring, which can improve offspring survival. There is fascinating diversity in the ways in which parents care for their offspring, from infant carrying behavior in titi monkeys, poison dart frogs and spiders to provisioning of offspring in burying beetles and birds[1,2]. The burden of parental care does not always land exclusively on females; in some species both parents provide care and in others males are solely responsible for care.

Our understanding of the molecular and neuroendocrine basis of parental care has been largely influenced by studies in mammals, where maternal care is the norm. In mammals, females experience cycles of estrus, pregnancy, child birth and lactation as they become mothers, all of which are coordinated by hormones. While maternal care is often primed by hormonal and physiological changes related to embryonic or fetal development, the primers for paternal behavior are likely to be more subtle, such as the presence of eggs or offspring[3,4]. Despite this subtlety, there is growing evidence that males can also experience changes in physiology and behavior as they become fathers, some of which resemble changes in mothers[5]. For example, men experience increased oxytocin[6] and a drop in testosterone[7] following the birth of a child. Indeed, a recent study in burying beetles showed that the neurogenomic state of fathers when they are the sole providers of care closely resembles the neurogenomic state of mothers[8].

There is taxonomic diversity in the specific behavioral manifestations of care, but all care-giving parents go through a predictable series of stages as they become mothers or fathers, from preparatory stages prior to fertilization (e.g. territory establishment and nest building) to the care of developing embryos (e.g. pregnancy, incubation), to care of free-living offspring (e.g. provisioning of nestlings, lactation, etc). Each stage is characterized by a set of behaviors and events, and the transition to the next stage depends on the successful completion of the preceding stage e.g. ref. [9]. The temporal ordering of stages, combined with our understanding of the neuroendocrine dynamics of reproduction[10], prompts at least three non-mutually exclusive hypotheses about how we might expect gene expression in the brain to change over the course of parental care. First, because each stage is characterized by a particular set of behaviors, each stage might have a unique neurogenomic state associated with it (the unique hypothesis). Second, some of the demands of parenting remain constant across stages, e.g. defending a nest site, therefore we might expect to see the signal of a preceding stage to persist into subsequent stages (the carryover hypothesis), resulting in shared genes among stages, especially between stages close together in the series. Finally, extending the reasoning further, and considering that parents must pass through one stage before proceeding to the next, genes associated with one stage might be added to the previous stage as a parent proceeds through the stages (the additivity hypothesis, an extension of the carryover hypothesis).Whether changes that occur at the neurogenomic level can be mapped on to behaviorally defined (as opposed to endogenously defined) stages of parental care is unknown. Moreover, we know little about whether there are genes that conform to a unique, carryover or additive pattern across stages of care. These hypotheses provide a novel conceptual framework for improving our understanding of parental care at the molecular level, and could serve as a model for studying other life events that comprise a series of behaviorally defined stages, e.g. stages of territory establishment, stages of pair-bonding, stages of dispersal, etc.

Unlike mammals, paternal care is relatively common in fishes: of the fishes that display parental care, 80% of them provide some form of male care, therefore fish are good subjects for understanding the molecular orchestrators of paternal care[11,12].

Moreover, the basic building blocks of parental care are ancient and deeply conserved in vertebrates[13]. For example, the hormone prolactin was named for its essential role in lactation in mammals, but had functions related to parental care in fishes long before mammals evolved[14]. Growing evidence for deep homology of brain circuits related to social behavior[15–18] suggests that the diversity of parental care among vertebrates is underlain by changes in functionally conserved genes operating within similar neural circuits[19].

In this study, we track the neurogenomic dynamics of the transition to fatherhood in male stickleback fish by measuring gene expression (RNA-Seq) in two brain regions containing nodes within the social behavior network, diencephalon and telencephalon. Gene expression in experimental males is compared across five different stages (nest, eggs and three time points after hatching) and relative to a control group. In this species, fathers are solely responsible for the care of the developing offspring, and male sticklebacks go through a predictable series of stages as they become fathers, from territory establishment and nest building to mating, caring for eggs, hatching and caring for fry[20].

In addition to providing care, parents must be vigilant to defend their vulnerable dependents from potential predators or other threats. Tradeoffs between parental care and territory defense have been particularly well studied in the ecological literature, e.g.[21], and parental care and territorial aggression represent the extremes on a continuum of social behavior—from strongly affiliative to strongly aversive. Therefore, an additional goal of this study is to compare and contrast the neurogenomics of paternal care with the neurogenomic response to a territorial challenge. As parental care and territorial aggression are social behaviors and both utilize circuitary within the social behavior network in the brain[15–17], we expect to observe similarities between parental care and a territorial challenge at the molecular level. However given their position at opposite ends of the continuum of social behavior, along with neuroendocrine tradeoffs between them[22], here we test the hypothesis that opposition between parental care and territorial aggression is reflected at the molecular and/or gene regulatory level.

Altogether results suggest that some of the molecular mechanisms of parental care are deeply conserved and are not sex-specific, and suggest that tradeoffs between opposing social behaviors are managed at the gene regulatory level.

## Results

**Neurogenomic dynamics of paternal care**. There were dramatic neurogenomic differences associated with paternal care. A large number of genes—almost 10% of the transcriptome—were differentially expressed between the control and experimental groups over the course of the parenting period (Fig. 1a, Supplementary Data 1). Within each stage, a comparable number of genes were up- and down-regulated. There were significant gene expression differences between the control and experimental groups within both brain regions; relatively more genes were differentially expressed in diencephalon.

Functional enrichment analysis of the differentially expressed genes (DEGs) suggests that paternal care requires changes in energy metabolism in the brain along with modifications of immune system and transcription. Genes associated with the immune response were down-regulated in both brain regions and during most stages relative to the control group. Genes associated with energy metabolism and the adaptive component of the immune response were upregulated in telencephalon. Genes associated with the stress response were downregulated in both brain regions around the day of hatching. Genes associated with energy metabolism were downregulated as the fry emerged

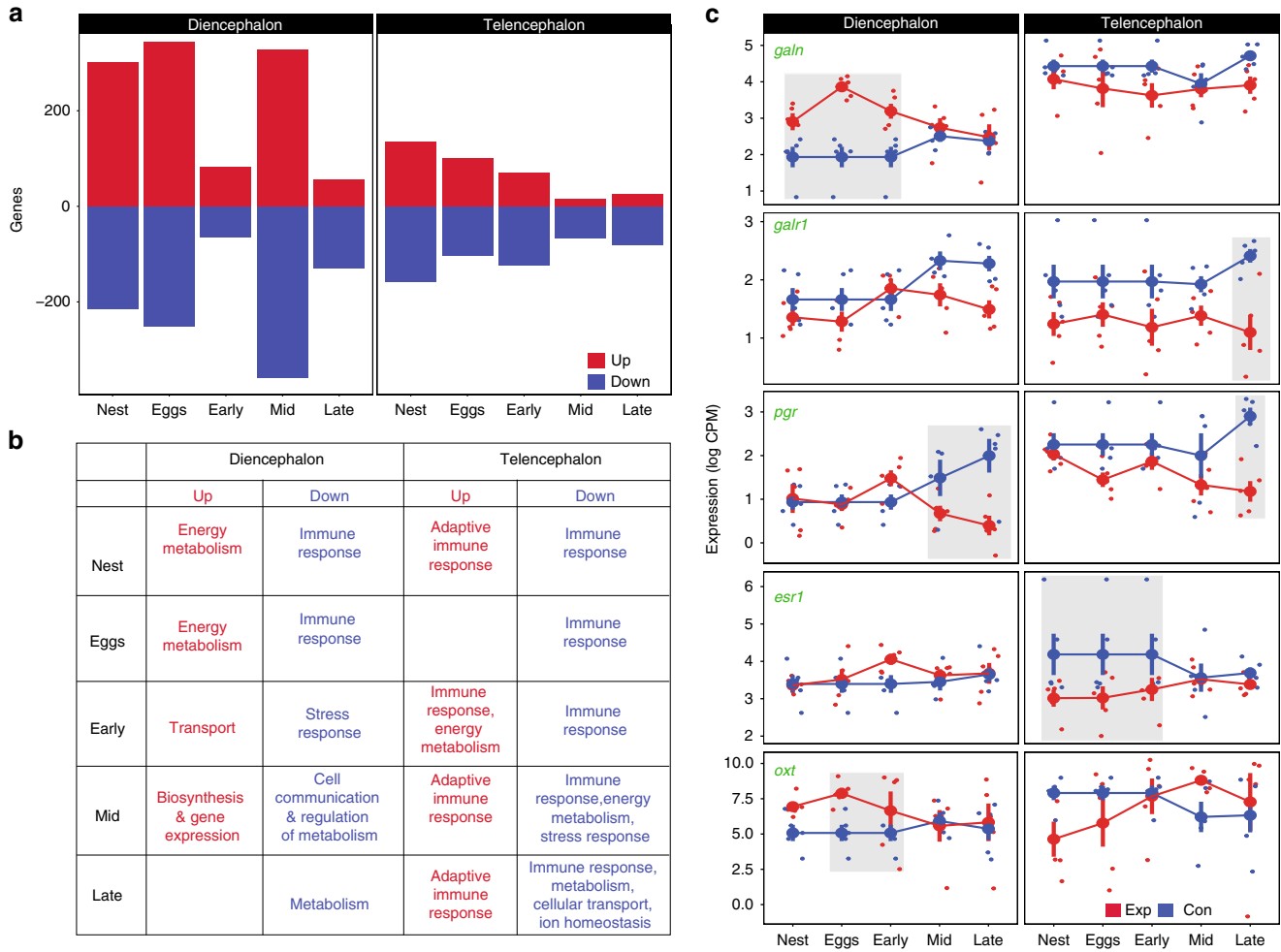

**Fig. 1** Neurogenomic dynamics of paternal care. **a** The number of up- and down-regulated differentially expressed genes (DEGs) at each stage of paternal care in diencephalon and telencephalon. **b** Summary of GO-terms that were enriched in up- and down-regulated genes at each stage in the two brain regions. **c** The expression profile of candidate genes related to maternal care (galanin, galanin receptor 1, progesterone, estrogen receptor 1, oxytocin) across stages, with expression in the two brain regions plotted relative to the appropriate circadian control group; data points represent individual samples with means and s.e.m. indicated. Statistical significance of these genes was assessed as a pairwise contrast between a stage and its control (see Supplementary Data 1 for full list of genes; source data are in GEO GSE134508) using negative binomial distribution with generalized linear models in edgeR. Boxes surround means that are statistically different between the control and experimental condition within the stage.

(Fig. 1b, Supplementary Data 2). The expression profile of particular candidate genes related to parental care are in Fig. 1c, with statistically significant differences between the control and experimental condition within a stage indicated. Altogether these patterns suggest that paternal care involves significant neurogenomic shifts in stickleback males.

**Change and stability of neurogenomic state across stages.** We used these data to assess evidence for three non-mutually exclusive hypotheses about how neurogenomic state might change across stages of parental care. According to the unique hypothesis, there is a strong effect of stage on brain gene expression and little to no overlap among the genes associated with different stages. To evaluate this hypothesis we tested whether there were DEGs that were unique to each stage, i.e. not shared with other stages. We generated lists of genes that were differentially expressed between the control and experimental group at each stage within each brain region. Then, we excluded the DEGs that were shared between stages in order to identify unique genes to each stage. To increase confidence that the unique genes are truly unique to each stage, i.e. that they didn't just barely passed the

cutoff for differential expression in another stage (false negatives), we followed an empirical approach (as in[23]). We kept the cutoff for DEGs at the focal stage at FDR < 0.01 and relaxed the FDR threshold on the other stages (Supplementary Fig. 1). This procedure was repeated for each stage and in each brain region separately. This analysis produced—with high statistical confidence—lists of DEGs that are unique to each stage (Fig. 2a), consistent with the "unique" hypothesis.

Next, we assessed the extent to which genes were shared among different stages of paternal care by testing whether the number of overlapping DEGs between stages was greater than expected using a hypergeometric test. Consistent with the carryover hypothesis, within each brain region, the number of overlapping DEGs between stages was statistically much greater than expected by chance (Supplementary Data 3), and stages that are close together in the series shared more DEGs compared to stages that are further apart (Fig. 2b, Supplementary Fig. 2).

These results suggest that there are genes whose signal persists across stages of care. We then evaluated the possibility that each new stage triggers a neurogenomic response which persists into subsequent stages, i.e. that genes associated with one stage are added to the previous stage as a parent proceeds through the

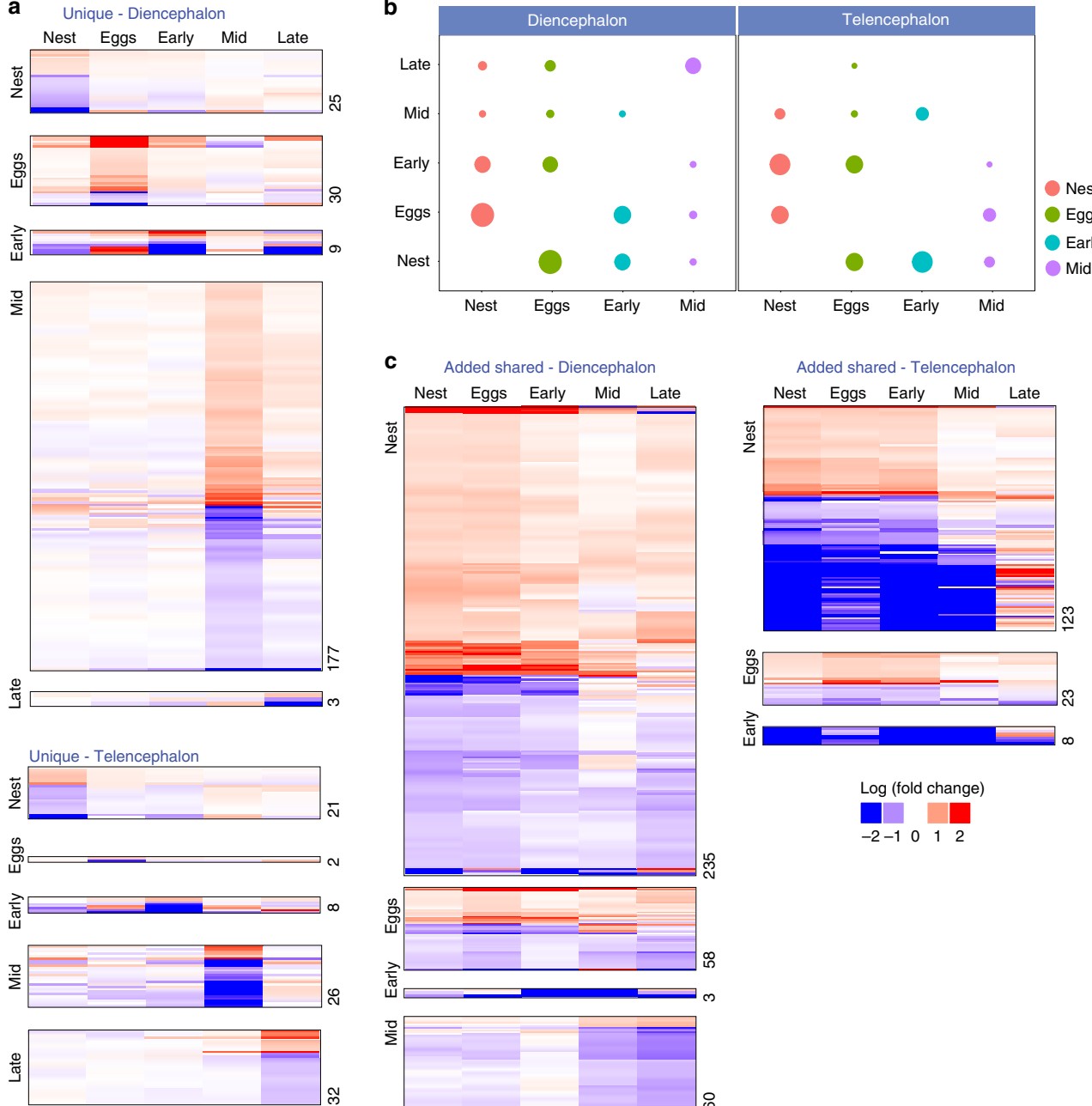

**Fig. 2** Change and stability of neurogenomic state across stages of parental care. **a** There were DEGs that were only differentially expressed during one stage. Shown is a heat map depiction of the expression profile of the genes that were "unique" to each stage, showing how they were regulated in other stages, separated by stage and by brain region. **b** The statistical significance of the pair-wise overlap between stages within each brain region. The size of the circle is proportional to the significance of the p-value (hypergeometric test FDR) of the overlap, such that large circles indicate smaller p-values. Note that the stages closest to the focal stage tended to share more DEGs compared to stages further apart in the series. **c** DEGs that were added to a stage and were also differentially expressed in subsequent stages. Shown is a heat map depiction of the added shared genes for each stage, separated by brain region, showing how they were regulated across stages. Red = upregulated, blue = downregulated. Numbers on the heat maps indicate the number of genes in each heat map. Source data are in GEO GSE134508

stages. According to this hypothesis, when a parent is caring for eggs in their nest, for example, the "egg" genes are added to the previously activated "nest" genes, and so on, in an additive fashion. To examine this statistically, for each stage, we identified genes that: (1) were differentially expressed during the stage of interest; (2) were not differentially expressed during any of the preceding stages; (3) were also differently expressed in a subsequent stage, hereafter referred to as "added shared genes". Only genes added during a new stage were used to test for their

overlap with subsequent stages, therefore except for the "nest added shared genes", each of the added shared genes from the previous stage(s) were subtracted from the focal stage's added shared genes (Supplementary Fig. 3). This process generated four sets of added shared genes: genes that were differentially expressed during the nest stage and were also differentially expressed during at least one subsequent stage ("nest added shared genes"), genes that were differentially expressed during the egg stage and were also differentially expressed during at least one

subsequent stage but not during the nest stage ("egg added shared genes"), and so on.

This analysis revealed genes that became differentially expressed as males proceeded through different stages of paternal care and ROAST[24] analysis found that the added shared genes remained differentially expressed in subsequent stages in a statistically significant manner (Supplementary Data 4). This suggests, for example, that there was a transcriptional signal of eggs which persisted after the egg stage. To see if the genes that were added and which persisted over time were similarly regulated across subsequent stages of paternal care, we examined the expression profiles of the added shared genes at each stage and tested if the direction of regulation was consistent across stages. This analysis revealed that added shared genes were indeed similarly regulated across stages (Supplementary Data 4, Fig. 2c). For example, added shared genes that were upregulated in males with nests were also upregulated during subsequent stages, especially during stages close to the nesting stage. To investigate this further, we calculated the probability that all genes within a set of added shared genes were expressed in the same direction due to chance, i.e. either consistently up- or down-regulated. Then, we counted the number of genes within each set of added shared genes that were concordantly expressed. We found that the number of concordantly expressed genes was greater than expected by chance (diencephalon $\chi^2 = 1859$, $P < 1e-6$, telencephalon $\chi^2 = 146$, df = 2, $P < 1e-4$). For example, 172 of the 235 genes in the nest added shared genes in diencephalon were concordantly expressed across stages, much higher than the expected 15 genes due to chance. The concordant expression pattern across stages suggests that an added shared gene serves a similar function in different stages.

**Pathways are not sex-specific and are deeply conserved.** Some of the candidate genes associated with female pregnancy and maternal care were differentially expressed in different stages of paternal care in sticklebacks (Fig. 1c). For example, in mammals, levels of progesterone, estrogen and their receptors increase during pregnancy and then subside after childbirth. A similar pattern was observed in the diencephalon of male sticklebacks: both estrogen receptor (esr1) and progesterone receptor (pgr) were upregulated during early hatching and then subsided (Fig. 1c). Oxytocin (and its teleost homolog isotocin) plays an important role in social affiliation and parental care in mammals[6] and fish[19,25–28]. Oxytocin (oxt) was upregulated in diencephalon when male sticklebacks were caring for eggs in their nests, and upregulated in telencephalon mid-way through the hatching process (Fig. 1c).

Genes that have been implicated in infanticide during parental care in mammals were also differentially expressed in sticklebacks, where egg cannibalism is common. Galanin—a gene implicated with infanticidal behavior in mice[29]—was highly expressed in diencephalon (which includes the preoptic area) during the nest, eggs and early hatching stages. However, the galanin receptor gene was downregulated during the middle to late hatching stages in both brain regions (Fig. 1c). Furthermore, the progesterone receptor—which mediates aggressive behavior toward pups in mice[30]—gradually declined in both brain regions as hatching progressed, and its level was lowest when all the fry were hatched (Fig. 1c). Up-regulation of galanin during the egg stage and down-regulation of progesterone receptor during the hatching stage could reflect how male sticklebacks inhibit cannibalistic behavior while providing care.

To test if the neurogenomic changes that we observed in stickleback fathers across stages, e.g. unique and added shared genes, are similar to the neurogenomic changes that mothers experience across stages of maternal care, we leveraged a recent dataset where brain gene expression was compared across a series of pregnancy and post-partum stages in mice (Supplementary Data 5)[31]. Similar to stickleback fathers, there were both unique and added shared DEGs across different stages of pregnancy and postpartum in mouse mothers. We then tested if the enduring (added shared genes) and transient (unique) changes in neurogenomic state that were experienced in stickleback fathers were similar to the enduring and transient signals of pregnancy and the postpartum period in mouse mothers. Specifically, we compared mouse and stickleback added shared genes within the appropriate orthogroup (Supplementary Data 6). For example, we compared 356 stickleback added shared genes within 90 orthogroups in diencephalon and 838 mouse added shared genes within 265 orthogroups in hypothalamus and found that they shared 14 orthogroups. In order to test whether those 14 shared orthogroups is greater than expected due to chance, we employed a Monte Carlo based permutation approach. We did not use a regular hypergeometric test or regular permutation test here (at the orthogroup level) because each orthogroup contains more than one gene in both the stickleback and mouse genomes, and some of those genes were differentially expressed and others were not. Instead, we sampled the gene sets (e.g. 356 and 838 genes in diencephalon/hypothalamus) repeatedly ($10^5$) and with replacement from both species' universes and counted the overlaps at the orthogroup level. This overlap was then tested against the observed overlap to compute p-values, which are highly significant (Fig. 3, note that the overlap never reaches 14 orthogroups). Added shared genes in stickleback and mouse include BDNF (a candidate gene related to anxiety, stress and depression[32]) and a regulator of G protein receptors RGS3 (related to insulin metabolism[33]). We followed the same procedure for the unique genes and did not find any evidence of sharing between the two species. For example, there were 33 unique genes in four orthogroups in mouse hypothalamus and 244 unique genes in 54 orthogroups stickleback diencephalon with no overlap between them (Supplementary Data 6).

Altogether, the differential expression of candidate genes related to maternal care along with the deep homology of the enduring signal of care across stages (added shared genes) suggest that some of the neurogenomic shifts that occur during paternal care in a fish are deeply conserved and are not sex-specific.

**Parenting and aggression tradeoffs at the molecular level.** To better understand how different social demands are resolved in the brain, we compared these data to a previous study on the neurogenomic response to a territorial challenge in male sticklebacks[34], which measured brain gene expression 30, 60 or 120 min after a 5 min territorial challenge. The two experiments studied behaviors at the opposite ends of a continuum of social behavior: paternal care provokes affiliative behavior while a territorial challenge provokes aggressive behavior, and the challenge hypothesis originally posited that patterns of testosterone secretion reflects tradeoffs between parental care and territory defense, assuming that testosterone is incompatible with parental care in males[22]. Subsequent studies have shown that testosterone is not always inhibitory of parental care[35], and that a territorial challenge activates gene regulatory pathways that do not depend on the action of testosterone[36]. Regardless of the specific neuromodulators or hormones, a mechanistic link between parental care and territory defense is likely to operate through the social behavior network in the brain because most nodes of this network express receptors for neuromodulators and hormones that are involved with both parental care and aggression[37]. Therefore we used these data to assess whether there is commonality at the molecular level between aggression and paternal care. For example,

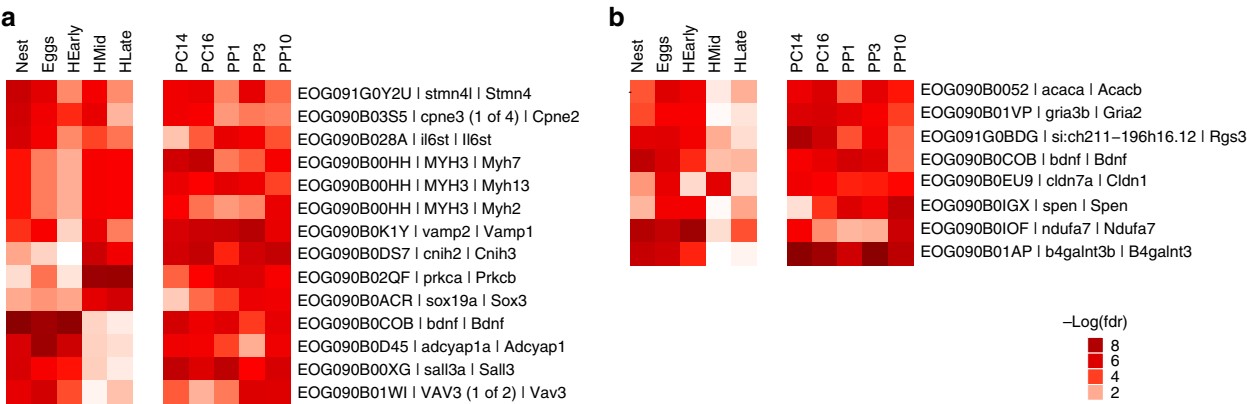

**Fig. 3** DEGs associated with shared orthogroups. Color represents the significance of differential expression between the control and experimental group (*p* values (−log(fdr)) across the five conditions in stickleback (left) and the five conditions in mouse (right). **a** shows the significance of DEGs within 14 shared orthogroups between diencephalon in stickleback and hypothalamus in mouse. **b** shows the significance of DEGs within nine shared orthogroups between telencephalon in stickleback and hippocampus in mouse. Source data are in GEO GSE134508

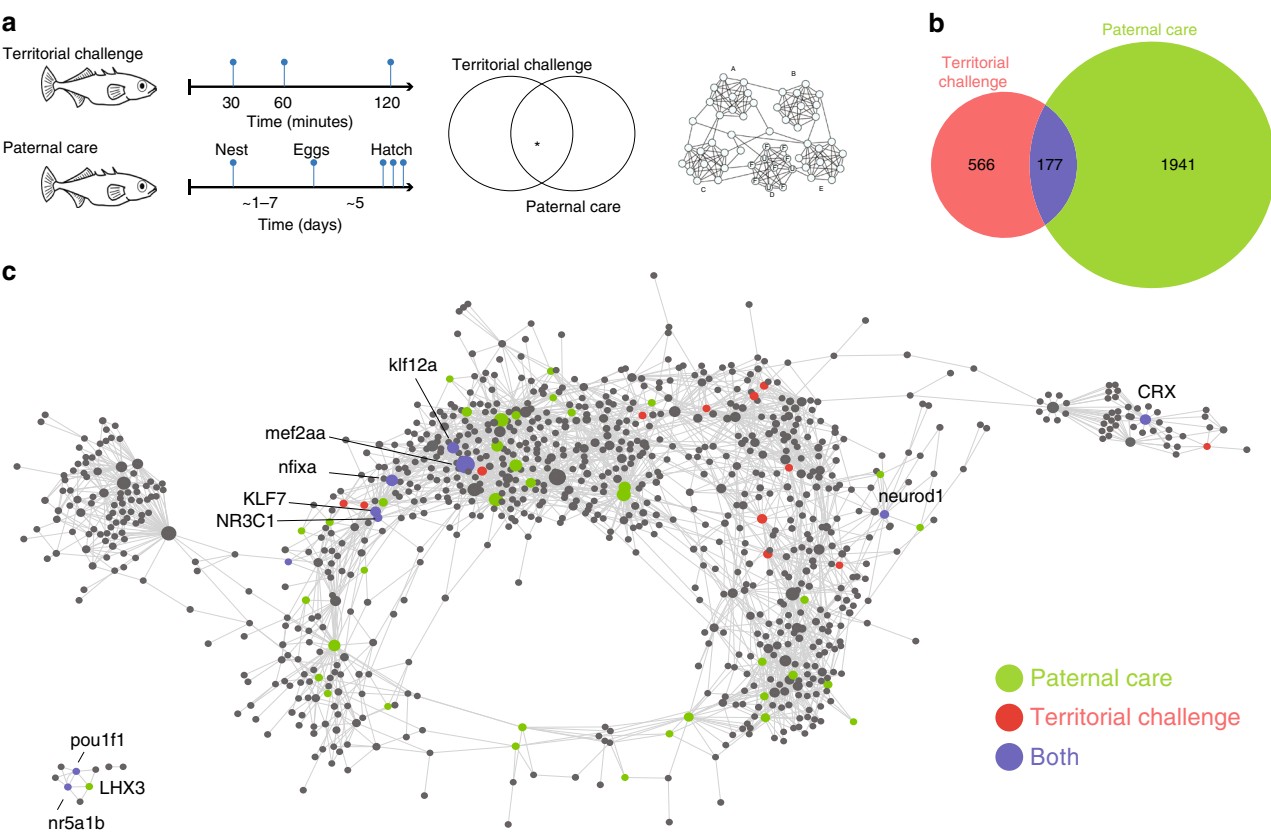

**Fig. 4** The regulatory dynamics of territorial challenge and paternal care. **a** Experimental time course sampling design in the two experiments. **b** Overlap between territorial aggression and paternal care DEGs. DEGs were pooled across time points and brain regions. **c** ASTRIX-generated transcriptional regulatory network. Each node represents a transcription factor or a predicted transcription factor target gene. Oversized nodes are transcription factors where the size of the node is proportional to the number of targets. Transcription factors whose targets are significantly enriched in either or both experiments are highlighted with different colors. Stickleback imaged drawn by MB. Source data are in GEO GSE134508

shared genes could reflect general processes such as the response to a social stimulus, while genes that are specific to an experiment could reflect the unique biology of paternal care versus territorial aggression. Alternatively, there might be a set of genes that is associated with both parental care and territorial aggression, but those genes are regulated in different ways depending on whether

the animal is responding to a positive (parental care) versus negative (territorial challenge) social stimulus.

To compare the neurogenomics of paternal care and the response to a territorial challenge at the gene level, we pooled genes that were differentially expressed in the experimental compared to the control group (FDR < 0.01) across time points,

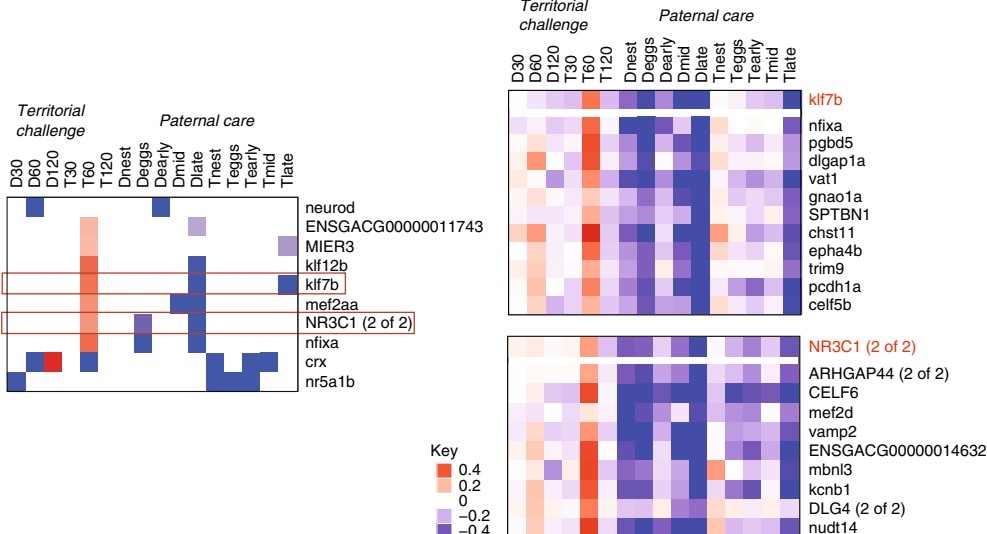

**Fig. 5** Shared regulators of a territorial challenge and paternal care. The panel on the left shows the expression pattern of the 10 transcription factors that were enriched in both experiments (Fig. 4). Columns are conditions within the two experiments (30, 60 or 120 min after a territorial challenge, the five stages of paternal care in diencephalon (D) or telencephalon (T)). Note that 8 of the shared transcription factors were regulated in opposite directions and in different brain regions in the two experiments. The two panels on the right show the expression pattern of two examples of shared, differentially regulated transcription factors (Klf7b and NR3C1) and their targets across all of the conditions. Source data are in GEO GSE134508

stages and brain regions within each experiment, which resulted in two sets of genes associated with either a territorial challenge or paternal care (Fig. 4a). There were 177 genes that were shared between the two experiments (Fig. 4b); this overlap is highly statistically significant (hypergeometric test, fdr < 1e-10).

To identify genes that were unique to each experiment while guarding against false positives, we adopted the same empirical approach as described above (Supplementary Fig. 1). There were 153 genes unique to territorial challenge and 764 genes unique to paternal care and these unique genes were enriched with non-overlapping functional categories (Supplementary Data 7). For example, some of the genes that were unique to a territorial challenge were related to sensory perception and tissue development, whereas some of the genes that were unique to paternal care were related to oxidative phosphorylation and energy metabolism, which might reflect the high metabolic needs of males as they are providing care[38].

The large number of genes that were differentially expressed both during paternal care and in response to a territorial challenge prompted us to test for evidence of their common regulation at the gene regulatory level. Therefore, we used the data from both experiments to build a transcriptional regulatory network and asked if there are transcription factors whose targets were significantly associated with the DEG sets from the paternal care experiment, the territorial challenge experiment or both experiments (Fig. 4, Supplementary Data 8). There were 10 transcription factors that were significantly enriched in both experiments. Eight out of 10 transcription factors were regulated in opposite directions in at least one of the conditions in the two experiments (Fig. 5). Two of the transcription factors that were regulated in opposite directions (NR3C1 and klf7b) have been implicated with social behavior in other studies (the glucocorticoid receptor NR3C1 and psychosocial stress during pregnancy[39]; klf7b and austim spectrum disorder[40]). These patterns suggest that for some genes, different salient experiences—providing paternal care and territorial aggression—trigger opposite gene regulatory responses.

Interestingly, the transcription factors showing the opposite expression pattern were differentially expressed in different brain regions in the two experiments. Specifically, shared transcription factors and their predicted targets were up-regulated in

telencephalon in response to a territorial challenge and down-regulated in diencephalon during parental care. These findings point to the molecular mechanisms by which transcription factors might differentially modulate the social behavior network[15–17] in the brain to manage conflicts between paternal care and territory defense.

## Discussion

While maternal care has long been recognized as an intense period when the maternal brain is reorganized[41,42], our results suggest that paternal care also involves significant neurogenomic shifts. Many of the neuroendocrine changes that are experienced by mammalian mothers are driven by endogenous cues during pregnancy, birth and lactation, and are required for fetal growth and development[31,43], with the neural circuits necessary for maternal care being primed by hormones during pregnancy and the post-partum periods[42]. Our results suggest that males can also experience dramatic neuromolecular changes as they become fathers, even in the absence of ovulation, parturition, postpartum events and lactation and their associated hormone dynamics[5]. We observed dramatic neurogenomic changes in males in response to cues for care that are exogenous (e.g. the presence of nesting material) and social (e.g. the presence of eggs or the hatching of fry). Such dramatic neurogenomic shifts associated with paternal care might be especially likely to occur in species when fathers are the sole providers of parental care, such as in sticklebacks. The effects might not be as strong in biparental systems where fathers contribute less. Consistent with this hypothesis, in burying beetles, when males were the sole providers of care, their brain gene expression profile was similar to mothers, but when they were biparental, fathers' neurogenomic state was less similar to mothers'[8].

A key challenge for care-giving parents is to defend their home and vulnerable offspring from threats, such as territorial intruders. Behavioral trade-offs between parental care and territory defense are well-documented[35] and work in this area has been influenced by the challenge hypothesis[22], which originally posited that androgens mediate the conflict between care and aggression. By comparing the neurogenomic dynamics of paternal care and the response to a territorial challenge, our work offers insights

into the gene regulatory mechanisms by which animals resolve these conflicting demands. Our results suggest that opposing social experiences acting over different time scales—providing paternal care over the course of weeks versus responding to a territorial challenge over the course of minutes to hours—trigger opposite gene regulatory responses. In particular, an analysis of the predicted gene regulatory network identified transcription factors that were significantly enriched both following paternal care and in response to a territorial challenge, and the majority of the transcription factors (and their targets) were regulated in opposite directions in the two experiments (Fig. 5).

While previous studies have explored circuit-level changes in the social behavior network in response to different social stimuli[15], our results point to the molecular basis of differential modulation of the social behavior network: the transcription factors showing the opposite expression pattern were differentially expressed in different brain regions in the two experiments. Specifically, shared transcription factors and their predicted targets were up-regulated in telencephalon in response to a territorial challenge and down-regulated in diencephalon during parental care. These findings suggest the molecular mechanisms by which transcription factors might differentially modulate the social behavior network[15–17] in the brain to manage conflicts between paternal care and territory defense. A similar pattern was observed at the transcriptomic (rather than gene regulatory) level when neurogenomic states were compared between territorial aggression and courtship in male threespined sticklebacks: some genes that were upregulated after a territorial challenge were downregulated after a courtship opportunity[44]. These results are also consistent with a detailed mechanistic study which showed that transcription factors play a role in setting up neural circuits to mediate opposing behaviors[45].

Altogether our analysis of changes in neurogenomic state across stages of paternal care offers support for all three hypotheses proposed. For example, consistent with the unique hypothesis, there were genes that were unique to each stage. Genes exhibiting transient, stage-specific differential expression might be involved in facilitating the next stage, priming and/or responding to a particular event or stimulus during that stage, e.g. the arrival of offspring. Whether genes that were unique to a particular stage and not differentially expressed in other stages are a cause of future behavior or consequence of past behavior is unknown. We also found support for the carryover and additivity hypotheses: elements of an acquired neurogenomic state persisted into subsequent stages, which suggests that the events and behaviors that characterize a particular stage of paternal care (e.g. finishing a nest, the arrival of eggs, hatching) trigger a neurogenomic state that persists, perhaps for as long as those events and behaviors continue. Genes whose expression persists across stages could be involved in maintaining the previous neurogenomic state, and/or reflect the constant demands of parenthood, e.g. the nest must be maintained across all stages of care.

Moreover, our results suggest that changes in neurogenomic state in a fathering fish might share commonalities at the molecular level with the neurogenomic changes associated with maternal care in a mammal. The number of orthologous genes that were shared across stages of maternal care in mice[31] and paternal care in sticklebacks was greater than expected due to chance. This suggests that the neurogenomic state that is maintained across pregnancy and the post partum period in mice, for example, at least partially resembles the neurogenomic state that is maintained while a male stickleback is caring for eggs and while the eggs are hatching. These results suggest that maternal and paternal care might share similarities at the molecular level, and this finding is consistent with other studies showing that parental males and females can use the same hormones and molecular mechanisms to activate the same pathways in the brain[46].

The finding of partial commonality between paternal care in a fish and maternal care in a mammal adds to the growing body of work showing that the underlying neural and molecular mechanisms related to parental care might have been repeatedly recruited during the evolution and diversification of parental care[19,47]. Indeed, our results suggest that so-called "pregnancy hormones" and added shared genes (for instance BDNF and G protein regulators, RGS3) might have been serving functions related to care giving long before the evolution of mammals, and that these mechanisms operate just as well in fathers as they do in mothers. These commonalities with maternal care in mammals suggest that the neurogenomic shifts that occur during paternal care in a fish might be deeply conserved and might not be sex-specific. Animals have been dealing with the problem of how to improve offspring survival (as well as avoiding filial cannibalism) for a long time; our results suggest that they have relied on ancient molecular substrates to solve it.

## Methods

**Sticklebacks.** In sticklebacks, paternal care is necessary for offspring survival and is influenced by prolactin[48], and the main androgen in fishes (11KT) does not inhibit paternal care in this species[49]. Paternal care in sticklebacks is costly both in terms of time and energy[38], infanticide and cannibalism are common[20], and males must be highly vigilant to challenges from predators and rival males while caring for their vulnerable offspring.

Adult males were collected from Putah Creek, CA, a freshwater population, in spring 2013, shipped to the University of Illinois where they were maintained in the lab on a 16:8 (L:D) photoperiod and at 18 °C in separate 9-l tanks. Males were provided with nesting material including algae, sand and gravel and were visually isolated from neighbors.

In order to track transcriptional dynamics associated with becoming a father, we sampled males for brain gene expression profiling at five different points during the reproductive cycle ($n = 5$ males per time point): nest, eggs, early hatching, middle hatching and late hatching (control: reproductively mature males with no nests). Males in the nest condition had a nest but had not yet mated. Males in the eggs condition were sampled four days after their eggs were fertilized. Because males in the eggs condition were sampled four days after mating, the transcriptomic effects of mating are likely to have attenuated by the time males were sampled at this stage. Hatching takes place over the course of the 5th day after fertilization, and a previous study found that brain activation as assessed by Egr-1 expression was highest while male sticklebacks were caring for fry as compared to males with nests or eggs[50]. In order to capture males' response to the new social stimulus of their fry (see[51]), we focused on three time points on the day of hatching, which capture the start of the hatching process (9 a.m.), when approximately half of the clutch is hatched (1 p.m.) and when all of the eggs have hatched (5 p.m.).

Males in the nest, eggs and early hatching conditions were sampled at 9 a.m., males in the mid-hatching condition were sampled at 1 p.m. and males in the late hatching condition were sampled at 5 p.m. Males in these conditions were compared to reproductively mature circadian-matched control males that did not have a nest ($n = 5$ males per control group). Wild-caught females from the same population were used as mothers. Males were quickly netted and sacrificed by decapitation within seconds. All methods were approved by the IACUC of the University of Illinois at Urbana-Champaign (#15077).

**RNA sequencing.** Heads were flash frozen in liquid nitrogen and the telencephalon and diencephalon were carefully dissected and placed individually in Eppendorf tubes containing 500 µL of TRIzol Reagent (Life Technologies). Total RNA was isolated immediately using TRIzol Reagent according to the manufacturer's recommendation and subsequently purified on columns with the RNeasy kit (QIAGEN). RNA was eluted in a total volume of 30 µL in RNase-free water. Samples were treated with DNase (QIAGEN) to remove genomic DNA during the extraction procedure. RNA quantity was assessed using a Nanodrop spectrophotometer (Thermo Scientific), and RNA quality was assessed using the Agilent Bioanalyzer 2100 (RIN 7.5–10); one sample was excluded because of low RNA quality. RNA was immediately stored at −80 °C until used in sequencing library preparation.

The RNAseq libraries were constructed with the TruSeq® Stranded mRNA HT (Illumina) using an ePMotion 5075 robot (Eppendorf). Libraries were quantified on a Qubit fluorometer, using the dsDNA High Sensitivity Assay Kit (Life Technologies), and library size was assessed on a Bioanalyzer High Sensitivity DNA chip (Agilent). Libraries were pooled and diluted to a final concentration of 10 nM. Final library pools were quantified using real-time PCR, using the Illumina compatible kit and standards (KAPA) by the W. M. Keck Center for Comparative and Functional Genomics at the Roy J. Carver Biotechnology Center (University of Illinois). Single-end sequencing was performed on an Illumina HiSeq 2500 instrument using a TruSeq SBS sequencing kit version 3 by the W. M. Keck Center

for Comparative and Functional Genomics at the Roy J. Carver Biotechnology Center (University of Illinois). The 79 libraries were sequenced on 27 lanes.

**RNA Seq informatics**. FASTQC version 0.11.3[52] was used to assess the quality of the reads. RNA-seq produced an average of 60 million reads per sample (Supplementary Data 9). We aligned reads to the *Gasterosteus aculeatus* reference genome (the repeat masked reference genome, Ensembl release 75), using TopHat (2.0.8)[53] and Bowtie (2.1.0)[54]. Results of the TopHat alignment were largely in agreement with results from HISAT2[55] (Supplementary Fig. 4). Reads were assigned to features according to the Ensembl release 75 gene annotation file (http://ftp.ensembl.org/pub/release-75/gtf/gasterosteus_aculeatus/). We used the default settings in all the programs unless otherwise noted.

**Defining DEGs**. HTSeq v0.6.1[56] read counts were generated for genes using stickleback genome annotation. Any reads that fell in multiple genes were excluded from the analysis. We included genes with at least 0.5 count per million (cpm) in at least five samples, resulting in 17,659 and 17,463 genes in diencephalon and telencephalon, respectively. Count data were TMM (trimmed mean of $M$-values) normalized in R using edgeR v3.16.5[57]. Samples separated cleanly by brain region on an MDS plot; we did not detect any outliers. To assess differential expression, pairwise comparisons between experimental and control conditions were made at each stage using appropriate circadian controls. Because the nest, eggs and early stages were all sampled at 9 a.m., their expression was compared relative to the same 9 a.m. control group.

Diencephalon and telencephalon were analyzed separately in edgeR v3.16.5. A tagwise dispersion estimate was used after computing common and trended dispersions. To call differential expression between treatment groups, a "glm" approach was used. We adjusted actual $p$-values via empirical FDR, where a null distribution of p-values was determined by permuting sample labels for 500 times for each tested contrast and a false discovery rate was estimated[58]. Similarities across stages of care was assessed using hypergeometric tests and PCA (Supplementary Fig. 2).

For a fair comparison between our study and Ray et al.[31], we reanalyzed the Ray et al., gene expression dataset by applying the same model, dispersion estimates and false discovery rate procedures.

**Unique genes**. One of the goals of this study was to identify genes that uniquely characterized a particular state, e.g. to a particular stage of paternal care, or to either the territorial challenge or the paternal care experiment. To address the possibility that putative unique genes barely passed the cutoff for differential expression in another state (false negatives), we adopted an empirical approach, as in ref. [23]. We kept the cutoff for DEGs at the focal state at FDR < 0.01 and relaxed the FDR cutoff on the other states (see Supplementary Fig. 1 for an explanation of this procedure). This procedure was repeated for each state and in each brain region separately.

**Added shared genes**. We wanted to know how many of the genes that were differentially expressed in one stage remained differently expressed in the subsequent stages (added shared genes). To find added shared genes, we first selected those stages which had significant pairwise overlap between them (FDR < 0.05, hypergeometric test). Only those genes were tested for overlap with subsequent stages; in order to qualify as an added shared genes for a particular stage, the gene could not be differentially expressed during a preceding stage and had to be differentially expressed during a subsequent stage, but not necessarily the stage immediately following that particular stage. Except for the first stage, each stage's genes were first subtracted from the previous stages' DEGs and then tested for overlap with subsequent stages (Supplementary Fig. 3).

To assess the significance of added shared genes, we used rotation gene set testing functionality (ROAST)[24] in the limma package[59]. ROAST can test whether any of the genes in a given set of added shared genes are differentially expressed in the specified contrast and also can test if they are consistently regulated. ROAST tests for three alternative hypotheses: "Up", tests whether the genes in the set tend to be up-regulated, "Down" tests whether the genes in the set tend to be down-regulated and "Mixed" tests whether the genes in the set tend to be differentially without regard for direction of regulation. Here we used directional ROAST (null hypothesis either Up or Down) and separated the added shared genes by their direction of regulation (up or down) in a focal stage and then tested for their significant differential expression and consistent direction in subsequent stages. We also complemented this analysis with a Chi-Square test to determine whether the number of genes within a given set of overlapping genes showing a concordant expression pattern is greater than expected due to chance.

**Stickleback and mouse orthrogroups**. To compare stickleback and mouse genes we generated a reliable orthogroup map using OrthoDB, v9.1[60]. This map contained both one-to-one, one to many and many to many orthology associations between stickleback and mouse genes. This map contains 3790 orthogroups which represent 4820 stickleback and 4894 mouse genes.

**Overlap significance**. We tested the significance of unique and added shared DEGs between stickleback and mouse at the orthogroup level. We used Monte Carlo repeated random sampling to determine if an observed orthogroup overlap between species was statistically significant at $P < 0.05$[61]. For example, suppose $t^*$ is the observed orthogroup overlap between the stickleback and the mouse gene lists and $n_1$ and $n_2$ are gene set sizes respectively. We repeatedly and randomly drew samples of size $n_1$ from the stickleback genome and samples of $n_2$ from the mouse genome for $M$ times ($M = 10^5$) with replacement and detected an overlap $t_i$ for each iteration of $M$ and computed an estimated p-value using the following equation,

$$estimate \ \wp = \frac{1 + \sum_{i=1}^{M} I(t_i \geq t^*)}{1 + M} \tag{1}$$

where $I(.)$ is an indicator function.

**Transcriptional regulatory network (TRN) analysis**. ASTRIX uses gene expression data to identify regulatory interactions between transcription factors and their target genes. A previous study validated ASTRIX-generated TF-target associations using data from ModENCODE, REDfly, and DROID databases[62]. The predicted targets of TFs were defined as those genes that share very high mutual information ($P < 10^{-6}$) with a TF, and can be predicted quantitatively with high accuracy (Root Mean Square Deviation (RMSD) < 0.33 i.e prediction error less than 1/3rd of each gene expression profile's standard deviation. The list of putative TFs in the stickleback genome was obtained from the Animal Transcription Factor Database. Given TFs and targets sets ASTRIX infers a genome-scale TRN model capable of making quantitative predictions about the expression levels of genes given the expression values of the transcription factors. The ASTRIX algorithm was previously used to infer a TRN models for honeybee, mouse and sticklebacks[34,62–64]. ASTRIX identified transcription factors that are central actors in regulating aggression, maturation and foraging behaviors in the honeybee brain[62].

Here we used ASTRIX to infer a joint gene regulatory network by combining gene expression profiles from a previous study on the transcriptomic response to a territorial challenge in male sticklebacks[34] with the data from this experiment. Combining the two datasets increased statistical power to help identify modules that are shared and unique to the two experiments. Transcription factors that are predicted to regulate DEGs in either experiment were determined according to whether they had a significant number of targets as assessed by a Bonferroni FDR-corrected hypergeometric test.

**Functional analysis**. We derived GO assignments, using protein family annotations from the database PANTHER[65]. Stickleback protein sequences were blasted against all genomes in the database (PANTHER 9.0 85 genomes). This procedure assigns proteins to PANTHER families on the basis of structural information as well as phylogenetic information. Genes were then annotated using GO information derived from the 85 sequenced genomes in the PANTHER database.

GO analysis were performed in R using TopGo v.2.16.0 and Fisher's exact test. A $p$-value cut off of <0.01 was used to select for significantly enriched functional terms wherever possible. We summarized the GO terms into larger and general categories to get a general overview of the underlying biology. Terms were grouped together if they were in a similar pathway and/or based on semantic similarity. GO enrichments along with their respective $p$-values are in Supplementary Data 2 and 7.

**Reporting summary**. Further information on research design is available in the Nature Research Reporting Summary linked to this article.

## Data availability
The datasets generated during and/or analysed during the current study are available in GEO accession code number GSE134508.

## Code availability
Codes are available on GitHub (https://github.com/bukhariabbas/stickleback-paternal-care). All other relevant data is available upon request.

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

## Acknowledgements

We thank Gene Robinson, Mark Hauber, Dave Zhao, Saurabh Sinha, Lisa Stubbs, Mikus Abolins-Abols and members of the Bell lab for comments on the paper. Bukhari was supported by a Dissertation Improvement Grant from the University of Illinois during the preparation of this paper. This material is based upon work supported by the National Science Foundation under Grant No. IOS 1121980, by the National Institutes of Health under award number 2R01GM082937-06A1 and by a grant from the Simons Foundation to L. Stubbs and Gene Robinson.

## Author contributions

S.A.B. contributed to study design, analyzed the data and wrote the first draft of the paper. M.S. contributed to study design and data analysis. N.J., M.B., L.R.S. and R.T. contributed to study design and collected the data. A.M.B. designed the study, contributed to data analysis and interpretation and edited the paper. All authors approved the final version of the paper.

## Additional information

**Competing interests:** The authors declare no competing interests.

