## [Peer Review File · Nature Communications]

Reviewers' comments:

Reviewer #1 (Remarks to the Author):

The authors describe the analysis of a very interesting and carefully designed RNA-Seq experiment on neurogenomic states associated with male parental care in sticklebacks. Fish are a particularly useful clade for understanding paternal care, as in many species, male territoriality and mating platform construction has advanced to explicit male care of eggs and fry.

I have no qualms with the experimental design, though I must admit that I am not an expert in stickleback behavior. The RNA-seq dataset is more than sufficiently powerful for the questions the authors wish to ask. The paper is also well-written, and although by necessity much of the analysis is descriptive, the authors have constructed some interesting hypotheses to explicitly test.

I do somewhat disagree with the authors in the categorization of different neurogenomic "stages". To be more precise, I do not disagree that these stages exist behaviourally, but rather that the discrete categorization used by the authors may obscure more continuous gene expression patterns. I suspect this is more of a semantic argument than a conceptual one, but worth thinking through, as it could help build more realistic transcriptome models.

Regardless of the semantics, the authors use this approach to test the three models they lay out in Fig 1 with regard to the neurogenomic development of parental care. They conclude that Model 3, and additive model, is the best fit to their data. This may be, but the support they present for Model 3 is rather thin, and consists of the presentation of data that could fit either Model 2 or Model 3, without any quantitative support (e.g. what proportion of the transcriptome follows these models). As a result, we are left with no idea how the authors decided that Model 3 is the best fit, as the different models are in no way statistically tested against the data. This section needs proper model testing with statistical tests to be robust. Alternatively, if model testing is not possible, PCA analysis of gene expression at different stages could be used to reveal similarity or differences among states.

Reviewer #2 (Remarks to the Author):

Overall comments:

Excellent and biologically neglected question. Very nice experimental design. I think this will make an excellent addition to the field after clarifying some points and giving some more thought to some of the analyses.

Major comments:

Figure 1. I think the models are overly simplistic to represent complete possibilities. There is no reason the green box in Model 3 could not disappear at Stage 4. Thinking of cohorts of DEGs as purely additive (Model 3) or complete turnover (Model 2) is not realistic or consistent with the current literature (Burying beetle, Mouse, Ants, Humans). I think there just needs to be a changing of language around this in that the presented model represent the most extreme possible strategies/trends. Model 1 is also not referred to the text, nor are any models name specified in figure caption. Linking names with their description directly will help readers navigate that large caption efficiently.

Carryover gene analysis. This seems like an interesting idea. However, I do not see a strong argument for your exact implementation. This can be done using gene set enrichment tests of one stage to the subsequent one. It was not clear to me why a new protocol was implemented when things exist to test these types of questions. I also do not understand why it was not restricted to only subsequent stages. A gene involved at Nest, not at eggs, and then again at Early (Fig3D) might not be doing same thing. It might be involved in the gene networks that control two different behaviours, but not truly represent a “pure” carryover gene. Or involved in two different brain sub-regions within the large brain regions sampled here. I am thinking of something like a neuropeptide as a model here that regulates many things (behaviour and then physiology or vice versa), which could be involved at several stages but not represent a carryover gene as described.

The visualisation was also hard to get through. I might just get rid of 3A. The Venn diagram is impossible to readily understand. The rest of figure is compact, but more visually interpretable. It also fools the reader to colour with a gradient. It is a binomial output; e.g., did a 0.01 FDR Nest gene make 0.01 FDR at each other stage, yes or no. Although it does help the reader understand polarity.

Line 205 – Why a FDR of 0.20, why not 0.10? How robust are these results to the secondary FDR cutoff? No citation or justification is given for this parameter value.

Paragraph 218 – Completely disagree. It is evidence there is a mixture of Model 2 & 3. Or if you do think it is 70% Model 3 vs 30% Model 2 and so it is really Model 3, numbers need to be put on. What are the percentages? It looks like there is a large, relative signal of “uniqueness” in the dien. and “carryover” in the tele.

Paragraph 225 – No results are given in this results section. There are no test statistics or p-values (or Bayes Factor) given for any candidate gene. The reader is only shown lines. Why make the reader guess which one has just enough variation not to make overall significance or which one has just main effects vs interactions?

Figure 4. An overlap of 14 orthogroups produced an FDR of <0.001? There were 2000 DEGs here and 3790 orthogroups. How many DEGs in the orthogroups for Ag and for Mm? Why sample with replacement for the overlap significance? Once a gene is chosen, it cannot be chosen again. That seems to lack biological meaning.

Line 368 – I did not see these results in the Results section. I did when I went back up, but I think these results need to be unpacked more.

Line 458 - Why were such old read aligning programs used? Both TopHat2 and Bowtie2 have been superseded by newer, and supposedly much improved, programs by the same groups (Salzberg/Landmead), HiSAT2.

Line 471 – Why were the two brain parts analysed separately? Why not analyse everything together and use contrasts to pick out specific comparisons with the model containing ID as a random effect? limma, much the same group as edgeR, has the ability to do this and is used here for other things.

Minor comments:

Line 42 – Any particular reason the updated version of the Evolution of Parental Care (2012) was not used?

Line 54. I know the BB people. There was an update of this work last year in Evolution showing more directly that females care genes are good predictors of male care genes.

Line 87 – Slightly misstated. Parental care is not common, but of the fishes that display parental care paternal care is the most common.

Line 99 – In this context, I would think aversion is the opposite of affiliative. I would argue aggression is an outcome of aversion, not its cause.

Paragraph 108 – It took me a long time to understand exactly what was sampled. Perhaps being more specific about this experimental design/sampling in this paragraph. I originally read territory, nest, mating, eggs (fanning), and hatching/caring (5 stages); and, your models have five stages. The actual sampling is nest, eggs, and three timepoints after hatching.

Line 141 – A comparable set within or across behavioural stages? Not clear after reading that twice.

Paragraph 144 – Specific results need to be added to support this paragraph. What were the top three enriched terms? The reader does not know if this is you scanning this list and “getting a feeling” of it or if there was a more formal analysis. Does Figure 2 include all top enriched terms for each stage? The reader just needs a better idea of how the summaries were done.

Paragraph 172 contains no results.

Figure 2C is not referred to in the text. Table S4 is the first SI Table referred to.

Line 229 – The gene names need to be listed before their abbreviation.

Line 439 – Why were two rounds of polyA selection used?

Line 452 – 50 samples (100 libraries) were sequenced over 27 lanes? How was that split up? What was the read count target?

Line 458 – Sample or Library? Split evenly, if by sample?

Line 461 – Was any general QC done on the samples to check for outliers? Huge number of libraries and lanes, hard not to imagine that there is not a flyer or two.

Line 464 – Why were multi-mapped reads excluded?

Line 458 – Why are both TopHat2 and Bowtie2 listed? TopHat2 is a wrapper and extension of Bowtie2.

Reviewer #3 (Remarks to the Author):

This interesting study by Bukhari et al examines gene expression in male stickleback as they are transitioning into different stages of fatherhood. The study also compares this stickleback gene expression to a maternal mouse data set and a stickleback territorial challenge data set to gain insight into commonalities in social behaviors.

Overall, the authors have made several interesting discoveries about genes underlying paternal care and their similarities/differences to mammalian maternal care and territorial behavior. The authors

raise interesting possibilities about the evolution of social/parental behaviors.

Throughout the manuscript there are places that need more detail, especially the contents of the supplemental tables, and how certain analyses were performed. Please see detailed criticisms below:

In Figure 2—how were the GO categories that are listed chosen? Are these the most significant, or was some other criteria used? The gene names are hard to read in C.

Figure 3—Did a carryover gene have to be expressed in the same direction—for example induced at the two stages or repressed at the two stages, relative to control? If not, then it does not seem to be a carryover gene, as the direction of expression changed. The authors should also make it clear that it is carryover of either induction or repression, if that is indeed the case.

In supplemental Table 7—Can the authors clarify what the comparisons are for each sheet. The authors indicate it is a control vs treatment, but should be more explicit.

In Supplemental Table 8—How are the carry over genes for a same-stage comparison defined. For example PC14 to PC14? What does PropUp and PropDown indicate? Can you be explicit about what Up and Down mean in this table. What is a mixed P value? How were the number of genes determined in the NGenes column?

For Figure 4—Color indicates significance. What is the comparison that was tested for mouse? I assume for stickleback it was the controls for each stage?

For supplemental table 9—why are there 16 genes on the sheet “Orthogroup_D_Hypothalamus”, but on the summary the overlap is 14?

Why did the authors only compare carryover and unique genes from the mouse and stickleback data sets? I wonder why they did not compare genes that are differentially expressed, without the carryover or unique criteria added?

For Figure 6—what are the eight genes. Is crx one of the eight? I don't think it is correct to call that one opposite directions. What does the (2 out of 2) mean in that figure?

For Supplemental Table 1, 2 and 3 the authors should provide the full gene lists, not just the <.01 set. What is the rationale for not providing the full data set? Please check all supplemental table descriptions to be sure there is enough detail for a reader to understand.

The supplemental tables all need better information about what is in each column. For example, what do these column headers mean:

overlap.pat.agg Unique_agg Unique_pat

All of the figure text is small and it made it hard to see what the authors were describing.

A lot of the paper is focused on distinguishing between the models and it appears that is the rationale for defining carryover genes. Is this the best approach to analyze these data sets? Are these models generally accepted in their field?

Thank you for the opportunity to revise our manuscript in response to the referees' constructive comments. All three referees noted that the paper makes a valuable contribution to the literature and offered a number of helpful suggestions for improvement. Below, we first address general comments that were made by all three referees about the presentation of different models of change in neurogenomic state as a function of paternal care. Then, we then provide a detailed response to each of the referees' comments.

OVERALL RESPONSE

All three referees raised questions about the way we presented the analysis of how neurogenomic state might be expected to change across stages of paternal care. We realize now that by proposing three distinct models, we framed the issues too simplistically. We thank the referees for encouraging us to think through the problem more carefully.

In response to their comments, we removed Figure 1, which presented the models as though they were mutually exclusive, and reframed this section in terms of ***non-mutually exclusive hypotheses*** about the ways neurogenomic state might change over the course of paternal care. We hypothesize that there are genes that are unique to each stage of care, genes that carryover across stages of care, and genes that are "added" as males proceed through the stages. By framing the problem in terms of nonmutually exclusive hypotheses, we no longer make claims about which model fits the data best. Instead, we have evidence for all three types of genes, and we think this framework and these results will help the field to refine our understanding of a complex and fascinating problem.

In the process of considering the referees' comments, we realized that we needed to better highlight the novelty of the study and our analytical approach. While these ideas draw from the rich literature on the endocrine changes associated with reproduction, this is really the first attempt to apply them in the context of neurogenomics. Indeed, although there are several papers comparing brain gene expression differences across stages of reproduction (e.g.¹⁻⁷), the studies to date have been primarily motivated by trying to understand the brain, and have focused on differences among brain areas rather than across the stages (not to mention that they are all focused on mothers!). In contrast, our study is motivated more by behavioral questions rather than questions about the brain, although obviously the two topics are related to each other. We now highlight novelty and significance of these hypotheses for our understanding of behavior more explicitly.

The referees raised specific issues about our discussion of carryovers and additivity. We address these issues in three ways: 1) we reformulated the hypotheses and clarify that the additivity hypothesis is an extension of the carryover hypothesis; 2) we expanded the ROAST analysis to include a specific test of whether overlapping genes are congruently expressed, and now explain this analysis in the main body of the MS rather than in the methods; 3) we supplemented #2 with an additional analysis showing that the number of concordantly expressed overlapping genes is much greater than expected due to chance.

DETAILED RESPONSES TO EACH OF THE REFEREES' COMMENTS

Reviewer #1 (Remarks to the Author):

The authors describe the analysis of a very interesting and carefully designed RNA-Seq experiment on neurogenomic states associated with male parental care in sticklebacks. Fish are a particularly useful clade for understanding paternal care, as in many species, male territoriality and mating platform construction has advanced to explicit male care of eggs and fry.

I have no qualms with the experimental design, though I must admit that I am not an expert in stickleback behavior. The RNA-seq dataset is more than sufficiently powerful for the questions the authors wish to ask. The paper is also well-written, and although by necessity much of the analysis is descriptive, the authors have constructed some interesting hypotheses to explicitly test.

I do somewhat disagree with the authors in the categorization of different neurogenomic “stages”. To be more precise, I do not disagree that these stages exist behaviourally, but rather that the discrete categorization used by the authors may obscure more continuous gene expression patterns. I suspect this is more of a semantic argument than a conceptual one, but worth thinking through, as it could help build more realistic transcriptome models.

Thank you for pointing this out. We chose to sample for transcriptional profiling at stages of paternal care that have discrete events and behaviors associated with them but you’re right: we don’t know if the same categories apply at the neurogenomic level, and this is one of the questions we wanted to answer in this study; this is now mentioned on lines 79-81.

Regardless of the semantics, the authors use this approach to test the three models they lay out in Fig 1 with regard to the neurogenomic development of parental care. They conclude that Model 3, and additive model, is the best fit to their data. This may be, but the support they present for Model 3 is rather thin, and consists of the presentation of data that could fit either Model 2 or Model 3, without any quantitative support (e.g. what proportion of the transcriptome follows these models). As a result, we are left with no idea how the authors decided that Model 3 is the best fit, as the different models are in no way statistically tested against the data. This section needs proper model testing with statistical tests to be robust. Alternatively, if model testing is not possible, PCA analysis of gene expression at different stages could be used to reveal similarity or differences among states.

Thank you for these comments. See “Overall response” above. We considered using PCA to compare across stages; the carryover hypothesis predicts, for example, that stages closer together in time would more closely resemble each other with respect to PC1 than stages further apart in time. While this approach has its advantages, it was not clear to us how PCA could be used to test predictions of the “Unique” hypothesis. We think that we have addressed this concern now that we have clarified the different hypotheses, expanded the analysis and more carefully described the analysis strategy.

Reviewer #2 (Remarks to the Author):

Overall comments:

Excellent and biologically neglected question. Very nice experimental design. I think this will make an excellent addition to the field after clarifying some points and giving some more thought to some of the analyses.

Thank you for the constructive feedback!

Major comments:

Figure 1. I think the models are overly simplistic to represent complete possibilities. There is no reason the green box in Model 3 could not disappear at Stage 4. Thinking of cohorts of DEGs as purely additive (Model 3) or complete turnover (Model 2) is not realistic or consistent with the current literature (Burying beetle, Mouse, Ants, Humans). I think there just needs to be a changing of language around this in that the presented model represent the most extreme possible strategies/trends. Model 1 is also not referred to the text, nor are any models name specified in figure caption. Linking names with their description directly will help readers navigate that large caption efficiently.

See “Overall response” above. Figure 1 has been removed.

Carryover gene analysis. This seems like an interesting idea. However, I do not see a strong argument for your exact implementation. This can be done using gene set enrichment tests of one stage to the subsequent one. It was not clear to me why a new protocol was implemented when things exist to test these types of questions.

See “Overall response” above.

I also do not understand why it was not restricted to only subsequent stages. A gene involved at Nest, not at eggs, and then again at Early (Fig3D) might not to be doing same thing. It might be involved in the gene networks that control two different behaviours, but not truly represent a “pure” carryover gene. Or involved in two different brain sub-regions within the large brain regions sampled here. I am thinking of something like a neuropeptide as a model here that regulates many things (behaviour and then physiology or vice versa), which could be involved at several stages but not represent a carryover gene as described.

This is a really good point – a gene that is DE at stages 1 and 3 but not at stage 2 could be doing very different things. In response to the referees’ comments, we no longer define “carryover genes” in this way, and now test the carryover hypothesis by asking whether there are more genes that are shared between stages than expected due to chance, and if stages closer in the series are more similar than stages further apart.

That being said, to test the additivity hypothesis, we generated sets of genes that overlapped between a focal stage and at least one other subsequent stage, i.e. we did not restrict the overlapping genes to consecutive stages. It is first worth noting that the majority of genes that were shared between a particular stage and a subsequent stage were DE in the next stage (Figure 2b). That being said, we chose to include any subsequent, rather than just consecutive, stages for two statistical reasons. First, by including genes that were DE in subsequent, not just consecutive, stages, we generate a slightly longer gene list, which gave us more power for later analyses, e.g. the orthogroup comparison with mouse. Second, we suspected that many of the overlapping genes that were statistically DE in subsequent, but not consecutive stages, barely passed the statistical threshold in the consecutive stage. The ROAST analysis indeed suggests statistically that the overlapping genes were DE and concordantly expressed

across multiple stages, more than expected. Therefore statistically the chance that many of our “overlapping” and “additive” genes represent the type of genes mentioned by referee #2 is quite low. The congruent expression patterns strongly suggest that the pattern the reviewer has in mind is highly unlikely; this is now mentioned on lines 209-10.

The visualisation was also hard to get through. I might just get rid of 3A. The Venn diagram is impossible to readily understand. The rest of figure is compact, but more visually interpretable. It also fools the reader to colour with a gradient. It is a binomial output; e.g., did a 0.01 FDR Nest gene make 0.01 FDR at each other stage, yes or no. Although it does help the reader understand polarity.

Thank you for the suggestion. We have removed the Venn diagram and now show the significance of the overlap between a focal stage and all of the other stages in Figure 2b.

Line 205 – Why a FDR of 0.20, why not 0.10? How robust are these results to the secondary FDR cutoff? No citation or justification is given for this parameter value.

We acknowledge that the choice of FDR is a bit arbitrary and we agree that we could have used a relaxed FDR of 0.1 in contrast to FDR of 0.2, since we are declaring DE at $FDR < 0.01$ instead of $FDR < 0.05$. We decided to use this conservative threshold after working with our other datasets in our group (Stein et al 2018, now cited in the MS), where simulations have led us to conclude that FDR 0.20 gives reasonable safeguard against genes which could have made the overlap between two sets.

Paragraph 218 – Completely disagree. It is evidence there is a mixture of Model 2 & 3. Or if you do think it is 70% Model 3 vs 30% Model 2 and so it is really Model 3, numbers need to be put on. What are the percentages? It looks like there is a large, relative signal of “uniqueness” in the dien. and “carryover” in the tele.

See “Overall response” above. We revised this section to emphasize that we have evidence for unique genes, carryover genes and genes which are added to a stage and which persist across subsequent stages. The hypergeometric tests show support for the carryover hypothesis in both brain regions; the greater number of DEGs in diencephalon compared to telencephalon makes it appear as though the unique signal is stronger in diencephalon but we see evidence for both the unique and carryover hypotheses in both brain regions.

Paragraph 225 – No results are given in this results section. There are no test statistics or p-values (or Bayes Factor) given for any candidate gene. The reader is only shown lines. Why make the reader guess which one has just enough variation not to make overall significance or which one has just main effects vs interactions?

Thank you for this suggestion. Significant differences are now indicated on what is now Figure 1 (formerly Figure 2).

Figure 4. An overlap of 14 orthogroups produced an FDR of <0.001 ? There were 2000 DEGs here and

3790 orthogroups. How many DEGs in the orthogroups for Ag and for Mm? Why sample with replacement for the overlap significance? Once a gene is chosen, it cannot be chosen again. That seems to lack biological meaning.

We couldn't use a regular hypergeometric test here to test for the significance of shared orthogroups between mouse and stickleback because usually an orthogroup contains more than one gene in both the stickleback and mouse genomes and not all such genes are differentially expressed. Therefore, we employed a Monte Carlo based permutation approach, where we sampled the gene sets repeatedly (10^5) and with replacement from both species' universes and counted the overlaps at the orthogroup level. This overlap was then tested against the observed overlap to compute p-values. Moreover, all the p values were corrected for multiple comparisons. We cannot sample without replacement for this permutation as we will run out of sets after a couple of thousand of iterations. Mouse DEGs (3570) were represented in 924 orthogroups, and sticklebacks DEGs (2118) were represented in 504 orthogroups.

Line 368 – I did not see these results in the Results section. I did when I went back up, but I think these results need to be unpacked more.

We draw more attention to this result on lines 291-7 of the Results section.

Line 458 - Why were such old read aligning programs used? Both TopHat2 and Bowtie2 have been superseded by newer, and supposedly much improved, programs by the same groups (Salzberg/Landmead), HiSAT2.

One of the goals of this paper is to test for commonalities and differences between paternal care and a previous study from the lab on territorial aggression (Bukhari et al 2017). Therefore in this paper we wanted to use the same pipeline and programs as Bukhari et al 2017 for the sake of consistency and reproducibility of DEG sets.

Line 471 – Why were the two brain parts analysed separately? Why not analyse everything together and use contrasts to pick out specific comparisons with the model containing ID as a random effect? limma, much the same group as edgeR, has the ability to do this and is used here for other things.

Thank you for this suggestion. We did, indeed, consider building a full model which included treatment, brain region, stage and the interactions among them, with individual as a random factor in limma. However, we elected to focus the analysis within each brain region for at least three reasons. First, we had a priori biological hypotheses we wanted to test about differences between treatments and stages, but we didn't have clear a priori hypotheses about brain regions, in part because we sampled heterogeneous brain regions with multiple brain areas within them. Second, analyzing the brain regions separately allowed us to keep the models simpler and more interpretable – it would have been hard to interpret 3 way interactions between treatment, stage and brain region, for example. Finally, one of the goals of this paper was to compare to the results from our previously published study (Bukhari et al., 2017), which also analyzed brain regions separately, for the same reasons. We elected to use the same analytical approach, the same versions of the programs, etc. for the sake of consistency and reproducibility.

Minor comments:

Line 42 – Any particular reason the updated version of the Evolution of Parental Care (2012) was not used?

We now cite the classic Clutton-Brock book as well as the more updated edited volume by Royle et al.

Line 54. I know the BB people. There was an update of this work last year in Evolution showing more directly that females care genes are good predictors of male care genes.

Interesting!

Line 87 – Slightly misstated. Parental care is not common, but of the fishes that display parental care paternal care is the most common.

Corrected.

Line 99 – In this context, I would think aversion is the opposite of affiliative. I would argue aggression is an outcome of aversion, not its cause.

Corrected.

Paragraph 108 – It took me a long time to understand exactly what was sampled. Perhaps being more specific about this experimental design/sampling in this paragraph. I originally read territory, nest, mating, eggs (fanning), and hatching/caring (5 stages); and, your models have five stages. The actual sampling is nest, eggs, and three timepoints after hatching.

Corrected.

Line 141 – A comparable set within or across behavioural stages? Not clear after reading that twice.

Thanks for pointing out that this is confusing. We reworded slightly to clarify that within a given stage, a comparable number of genes were up and down regulated.

Paragraph 144 – Specific results need to be added to support this paragraph. What were the top three enriched terms? The reader does not know if this is you scanning this list and “getting a feeling” of it or if there was a more formal analysis. Does Figure 2 include all top enriched terms for each stage? The reader just needs a better idea of how the summaries were done.

We summarized the GO terms into larger and general categories to get a general overview of the underlying biology. Terms were grouped together if they were in a similar pathway and/or based on

semantic similarity. This has now been clarified on lines 535-8. GO enrichments along with their respective p values are in Table S2.

Paragraph 172 contains no results.

This paragraph was meant to be a transition to help the reader follow the reasoning of the analysis. If the referee and editor prefer, we can combine the next paragraph with the one in question (although the combined paragraph is pretty long).

Figure 2C is not referred to in the text.

Corrected.

Table S4 is the first SI Table referred to.

We now refer to each of the Supplementary Tables in the text.

Line 229 – The gene names need to be listed before their abbreviation.

Corrected.

Line 439 – Why were two rounds of polyA selection used?

Thank you for catching this error; it has been corrected.

Line 452 – 50 samples (100 libraries) were sequenced over 27 lanes? How was that split up? What was the read count target?

We now mention line 426 that 79 libraries were sequenced on 27 lanes, with 2-3 libraries per lane. This information is also provided in Table S9.

Line 458 – Sample or Library? Split evenly, if by sample?

Each lane sequenced 2-3 libraries (Table S9).

Line 461 – Was any general QC done on the samples to check for outliers? Huge number of libraries and lanes, hard not to imagine that there is not a flyer or two.

We did QC and did not find any outliers, both brain regions were separated on the mds scales. This is now mentioned on line 441-2.

Line 464 – Why were multi-mapped reads excluded?

Multi-mapped reads were excluded because it's difficult to know where these sequences came from, and those could cause over representation of feature abundances. In addition, multi-mapped reads could violate assumptions necessary for differential expression, where information is borrowed across features to improve on dispersion estimates.

Line 458 – Why are both TopHat2 and Bowtie2 listed? TopHat2 is a wrapper and extension of Bowtie2.

Since both programs depend on each other, we cited them separately.

Reviewer #3 (Remarks to the Author):

This interesting study by Bukhari et al examines gene expression in male stickleback as they are transitioning into different stages of fatherhood. The study also compares this stickleback gene expression to a maternal mouse data set and a stickleback territorial challenge data set to gain insight into commonalities in social behaviors.

Overall, the authors have made several interesting discoveries about genes underlying paternal care and their similarities/differences to mammalian maternal care and territorial behavior. The authors raise interesting possibilities about the evolution of social/parental behaviors.

Thank you for the positive feedback and helpful suggestions!

Throughout the manuscript there are places that need more detail, especially the contents of the supplemental tables, and how certain analyses were performed. Please see detailed criticisms below:

In Figure 2—how were the GO categories that are listed chosen? Are these the most significant, or was some other criteria used? The gene names are hard to read in C.

We summarized the GO terms into larger and general categories to get a general overview of the underlying biology. Terms were grouped together if they were in a similar pathway and/or based on semantic similarity. GO enrichments along with their respective p-values are in Table S2.

Figure 3—Did a carryover gene have to be expressed in the same direction—for example induced at the two stages or repressed at the two stages, relative to control? If not, then it does not seem to be a carryover gene, as the direction of expression changed. The authors should also make it clear that it is carryover of either induction or repression, if that is indeed the case.

Thank you for this thoughtful comment. We now perform two statistical tests to determine whether genes that overlapped between stages were indeed concordantly regulated, i.e. consistently up- or down-regulated in different stages. Both methods show that the overlapping genes are much more concordantly expressed than expected.

In supplemental Table 7—Can the authors clarify what the comparisons are for each sheet. The authors indicate it is a control vs treatment, but should be more explicit.

Thank you for pointing this out. Table S7 is now Table S6. We now provide more explanation about each sheet.

In Supplemental Table 8—How are the carry over genes for a same-stage comparison defined. For example PC14 to PC14? What does PropUp and PropDown indicate? Can you be explicit about what Up and Down mean in this table. What is a mixed P value? How were the number of genes determined in the NGenes column?

Table S8 is now Table S4. The columns are now defined in the table caption. We originally compared stages to themselves, e.g. PC14 to PC14, to test the functionality of the program (and found that it was working as expected); those comparisons have been removed.

For Figure 4—Color indicates significance. What is the comparison that was tested for mouse? I assume for stickleback it was the controls for each stage?

The reproductive female mice were compared to virgin females. For stickleback there were three circadian controls, which were reproductive males sampled at 9am, 1pm or 5pm. The nest, eggs and early stages were compared to the 9am control, the mid stage was compared to the 1pm control and the late stage was compared to the 5pm control.

For supplemental table 9—why are there 16 genes on the sheet “Orthogroup_D_Hypothalamus”, but on the summary the overlap is 14?

The overlap was tested at the orthogroup level; there were 16 DEGs and 14 orthogroups for this comparison.

Why did the authors only compare carryover and unique genes from the mouse and stickleback data sets? I wonder why they did not compare genes that are differentially expressed, without the carryover or unique criteria added?

Good question. We considered running two analyses: 1) comparing DEGs between species; 2) comparing carryover and unique genes between species. We decided for the sake of efficiency to focus on #2 because we had an *a priori* hypothesis about how we might expect to observe commonality between paternal and maternal care that was motivated by the stickleback results. Also, we reasoned that if we

detected commonality of carryover and unique genes between stickleback and mouse, that would by extension address question #1.

For Figure 6—what are the eight genes. Is crx one of the eight? I don't think it is correct to call that one opposite directions.

Crx is one of the eight genes. We see your point that Crx was differentially expressed three different conditions in the Territorial Challenge experiment, but was only expressed in the opposing direction relative to the paternal care experiment in one of the conditions (D120). We now qualify our inference on line 285. It is important to note, though, that we don't think the upregulation of Crx in D120 is an error, as its targets are similarly upregulated in the D120 condition as well (see below). White spaces in the heatmap indicates that the gene was not differentially expressed in the paternal care experiment.

What does the (2 out of 2) mean in that figure?

2/2 refers to the different paralog; teleost genomes have undergone genome duplication events.

For Supplemental Table 1, 2 and 3 the authors should provide the full gene lists, not just the <.01 set. What is the rationale for not providing the full data set? Please check all supplemental table descriptions to be sure there is enough detail for a reader to understand.

Thanks for the suggestion. We now provide the full gene list in Table S1 and added more description of the supplemental tables.

The supplemental tables all need better information about what is in each column. For example, what do these column headers mean:

overlap.pat.agg Unique_agg Unique_pat

We relabeled the columns and rewrote the figure caption to clarify.

All of the figure text is small and it made it hard to see what the authors were describing.

Sorry about that; this has been corrected.

A lot of the paper is focused on distinguishing between the models and it appears that is the rationale for defining carryover genes. Is this the best approach to analyze these data sets? Are these models generally accepted in their field?

See Overall Response, above. We thank the referee for this helpful comment; it helped us to refine our thinking about the novelty of the study, which is now mentioned explicitly on lines 83-6.

References cited

- 1 Akbari, E. M. *et al.* The effects of parity and maternal behavior on gene expression in the medial preoptic area and the medial amygdala in postpartum and virgin female rats: A microarray study. *Behavioral Neuroscience* **127**, 913-922, doi:10.1037/a0034884 (2013).
- 2 DiCarlo, L. M., Vied, C. & Nowakowski, R. S. The stability of the transcriptome during the estrous cycle in four regions of the mouse brain. *The Journal of Comparative Neurology* **525**, 3360-3387, doi:10.1002/cne.24282 (2017).
- 3 Eisinger, B. E., Zhao, C., Driessen, T. M., Saul, M. C. & Gammie, S. C. Large scale expression changes of genes related to neuronal signaling and developmental processes found in lateral septum of postpartum outbred mice. *PLoS One* **8**, e63824, doi:10.1371/journal.pone.0063824 (2013).
- 4 Gammie, S. C., Driessen, T. M., Zhao, C., Saul, M. C. & Eisinger, B. E. Genetic and neuroendocrine regulation of the postpartum brain. *Front Neuroendocrinol* **42**, 1-17, doi:10.1016/j.yfrne.2016.05.002 (2016).
- 5 Saul, M. C., Zhao, C., Driessen, T. M., Eisinger, B. E. & Gammie, S. C. MicroRNA expression is altered in lateral septum across reproductive stages. *Neuroscience* **312**, 130-140, doi:10.1016/j.neuroscience.2015.11.019 (2016).
- 6 Xiao, X. Q., Grove, K. L., Lau, S. Y., McWeeney, S. & Smith, M. S. Deoxyribonucleic acid microarray analysis of gene expression pattern in the arcuate nucleus/ventromedial nucleus of hypothalamus during lactation. *Endocrinology* **146**, 4391-4398, doi:10.1210/en.2005-0561 (2005).

- 7 Zhao, C., Saul, M. C., Driessen, T. & Gammie, S. C. Gene expression changes in the septum: possible implications for microRNAs in sculpting the maternal brain. *PLoS One* **7**, e38602, doi:10.1371/journal.pone.0038602 (2012).

Reviewers' comments:

Reviewer #1 (Remarks to the Author):

I really do like this dataset, and I appreciate that the authors have relaxed their rather strict hypotheses.

However, my concern remains that the authors have constructed a series of hypotheses, filtered their gene expression data explicitly based on these hypotheses, and found some genes that fit them. Apologies if I was not sufficiently articulate in the previous review as to why this is important. I'll give it another try here.

Transcriptome data are extremely rich, and it is possible to find genes that follow nearly any imaginable gene expression pattern depending on how the data are filtered. Without some more agnostic analysis, it really is not possible to know whether the global expression data fit the hypotheses presented, or whether there are just a few genes within the dataset that are consistent.

If the authors want to test their theories, they must show overall support between their entire transcriptome datasets and their specific hypotheses. The way it is reported now, they simply show that some genes are consistent, but (again) it is possible in transcriptome datasets to find genes with expression patterns consistent with nearly every possible idea. To take it an absurd level, I could hypothesize that gene regulation has nothing to do with the parental care cycle, that care is constitutively programmed. Filtering the data for genes that do not change over the course of parental, I would no doubt find a list of genes that fit this hypotheses. I could then do downstream analysis, such as GO, and perhaps find terms consistent with this idea (such as homeostasis, etc). But I would not know with this sort of analysis whether the entire dataset is a better fit to this hypotheses than alternatives. I simply report a series of genes consistent with the idea. This is precisely what the authors have done here. This is not, at least by my understanding, hypothesis testing.

This is why I originally suggested a PCA of all the transcriptome data. The similarity of given stages should be evident from the PCA, with temporally proximal stages in theory spatially proximate in the PCA, or even partially overlapping. PCA is quite commonly done with these type of data -for example, see Olivas et al. (New Phytologist 2016), Martinez-Abadias (Systematic Biology 2016), Bloch et al. (Nature Ecology & Evolution 2018) among many others.

PCA is only one option. The authors could use hierarchical clustering of their whole transcriptome data. Again, based on their predictions, we would expect temporally adjacent samples to show more similar expression than those at larger time distances (see Chanderbali et al PNAS 2010 for a nice example). Bootstrap values are quite useful in this type of analysis for showing overall support for transcriptional similarity.

Regardless of how they do it, I maintain that it is necessary for the authors actually test their hypotheses, rather than showing just a few cherry-picked loci that are consistent with them.

Minor comments.

Line 273. "however we used a more stringent FDR cutoff of $FDR < 0.25$ here instead of $FDR < 0.20$ ". Two questions. 1 – Why were the different cutoffs used for the same type of analysis? For consistency, the same one should be used in both. 2 – I think $FDR < 0.25$ is actually less stringent than $FDR < 0.2$, unless I misunderstand the nature of FDR corrections.

Line 443. "We included genes with at least one counts per million (cpm) in at least two samples". Two things. 1 – I think this should be "one count per million". 2 - Apologies for not noticing in the previous

draft, but this is a somewhat problematic threshold as it does not correct for differences in gene length. Because it is based on strict read count, longer genes will have a higher likelihood of passing than shorter genes as they will simply have more mappable reads overall. The authors should use a method that corrects for gene length such as RPKM, TPM.

Line 490. Change to Orthogroups.

Reviewer #2 (Remarks to the Author):

I like this experiment more every time I read the manuscript.

There are a few previous Issues raised that I do not think were addressed adequately. These were not strong counter arguments, in what otherwise was a strong rebuttal:

Original - Figure 4. An overlap of 14 orthogroups produced an FDR of <0.001 ? There were 2000 DEGs here and 3790 orthogroups. How many DEGs in the orthogroups for Ga and for Mm?

The test you describe is exact how I envisioned you did it. This just needs to be cleared up for me a bit more because it still seems unbelievable. From a list of 3790 elements, you sampled 904 elements (representing the Mm orthogroups that contain DEGs) and, independently, 504 elements (representing the Ga orthogroups that contain DEGs). You then asked for the mean intersection of these two list from 10,000 independent draws?

I do this 10 times and I get a mean of 117 +/- 9 elements. That suggests that 14 in the observed list is evidence of an underrepresentation of shared orthologs, rather than an enrichment. Is this what is being done? Unless the "genomes" mentioned are not just restricted to gene represented in the orthogroups, but all genes for each organism. Looking at the equation again, should you not be asking how many time t^* is less than, rather than greater than, t_i for a test of overrepresentation?

This seems to be a crux of the viability of Ga to be representative of vertebrates generally for parental care. I think this argument and its exact test needs to be fleshed out more for the reader.

Original - Line 458 - Why were such old read aligning programs used? Both TopHat2 and Bowtie2 have been superseded by newer, and supposedly much improved, programs by the same groups (Salzberg/Landmead), HISAT2.

That you want to make this analysis comparable to an older paper is ok. That is, however, an argument for reanalysing the old data with the newer methods and not an argument for analysing new data with a dated method. I was thinking doing a simple comparison between mapping rates of the two programs and showing no real difference exist. Otherwise, there is no need to produce new methods that address known problems with older methods or keeping abreast of best practices within a field.

Original - Line 471 – Why were the two brain parts analysed separately? Why not analyse everything together and use contrasts to pick out specific comparisons with the model containing ID as a random effect? limma, much the same group as edgeR, has the ability to do this and is used here for other things.

Wanting to test specific a priori hypotheses (aka, contrasts) within a model is not exclusive of running a full model. There is no need to talk about a 3 way interaction even if modelled, which I agree would be nigh impossible to interpret. Unless it is the only significant factor and even then it tells the reader

that there are not strong main effects. Having all samples from the same individual in one model allows the variance to be partitioned better, certainly for the random effect of individual. That you did it this way before is a reason to reanalyse old data with a more complete model and not a reason to analyse new data with a less suitable model.

Original - Line 464 – Why were multi-mapped reads excluded?

No. One does not need to know where a read exactly comes from for DE analysis. Using “compatibility counts” for each read/transcript comparison is the entire basis of the next generation of pseudo-alignment DE programs (Salmon, Kallisto), which are performing at the level of or better than older DE programs. Not a major issue, just again something where this paper is not aligned with the leading edge of RNA-seq. I do not think it biases the analyses, just limits their power.

Minor comments

Figure 1C is not mentioned in results in the “Figure 1” paragraph. I think it might worth mentioning that all the genes of Figure 1C had overall significance and that the stars for significance are not just a huge number of pairwise comparisons.

Reviewer #3 (Remarks to the Author):

Overall, I think the analysis and presentation are improved. The authors satisfactorily responded to all of my comments in the first review.

Additional comments:

While the authors suggest that the changes are “analogous to the changes associated with pregnancy, parturition and the postpartum period in mammalian mothers.”, the actual number of genes with overlap is small, even though the overlap is significant. However, the sets of genes with orthologs appears to be small based on STable 6 and the methods. Perhaps the authors can be more straightforward in the description of these results and lay out for the reader in the results the number of genes with orthologs that can actually be tested in that comparison. I also wonder what cut-off was used to consider a gene a gene an ortholog (p value, e value) and if it was too stringent. While all the information is in the manuscript, a casual reader may not realize that the number of genes with overlap are small and this may be because the number of orthologs called is small. This is especially important given that the observation regarding overlap with mouse/mammal maternal expression is highlighted in both the abstract and discussion of the manuscript, as an important discovery.

One minor comment is regarding the following sentence: “even in the absence of parturition, postpartum ovulation and lactation and their associated hormone dynamics”.

Do the authors mean “postpartum ovulation”? I wonder if that sentence is meant to indicate postpartum events, other than ovulation, that proceed lactation?

Our responses to the reviewers' comments are in red.

Reviewers' comments:

Reviewer #1 (Remarks to the Author):

I really do like this dataset, and I appreciate that the authors have relaxed their rather strict hypotheses. However, my concern remains that the authors have constructed a series of hypotheses, filtered their gene expression data explicitly based on these hypotheses, and found some genes that fit them. Apologies if I was not sufficiently articulate in the previous review as to why this is important. I'll give it another try here.

Transcriptome data are extremely rich, and it is possible to find genes that follow nearly any imaginable gene expression pattern depending on how the data are filtered. Without some more agnostic analysis, it really is not possible to know whether the global expression data fit the hypotheses presented, or whether there are just a few genes within the dataset that are consistent.

If the authors want to test their theories, they must show overall support between their entire transcriptome datasets and their specific hypotheses. The way it is reported now, they simply show that some genes are consistent, but (again) it is possible in transcriptome datasets to find genes with expression patterns consistent with nearly every possible idea. To take it an absurd level, I could hypothesize that gene regulation has nothing to do with the parental care cycle, that care is constitutively programmed. Filtering the data for genes that do not change over the course of parental, I would no doubt find a list of genes that fit this hypothesis. I could then do downstream analysis, such as GO, and perhaps find terms consistent with this idea (such as homeostasis, etc). But I would not know with this sort of analysis whether the entire dataset is a better fit to this hypothesis than alternatives. I simply report a series of genes consistent with the idea. This is precisely what the authors have done here. This is not, at least by my understanding, hypothesis testing.

This is why I originally suggested a PCA of all the transcriptome data. The similarity of given stages should be evident from the PCA, with temporally proximal stages in theory spatially proximate in the PCA, or even partially overlapping. PCA is quite commonly done with these type of data -for example, see Olivas et al. (New Phytologist 2016), Martinez-Abadias (Systematic Biology 2016), Bloch et al. (Nature Ecology & Evolution 2018) among many others.

PCA is only one option. The authors could use hierarchical clustering of their whole transcriptome data. Again, based on their predictions, we would expect temporally adjacent samples to show more similar expression than those at larger time distances (see Chanderali et al PNAS 2010 for a nice example). Bootstrap values are quite useful in this type of analysis for showing overall support for transcriptional similarity.

Regardless of how they do it, I maintain that it is necessary for the authors actually test their hypotheses, rather than showing just a few cherry-picked loci that are consistent with them.

This is a really thoughtful comment and thank you for explaining it more fully. We certainly agree that transcriptomic data are exceedingly rich and that one could, indeed,

cherry pick genes to fit any pattern we're looking for without doing statistics. We also agree that PCA and hierarchical clustering could be used to test the carryover hypothesis. Indeed, when we performed hierarchical clustering as the reviewer suggested, we found that stages close together in time are more similar (Figure 1). We agree this is a useful way to test the carryover hypothesis, and this figure has been added as Supplementary Figure S2.

Figure 1. Hierarchical clustering of logFC of all of the differentially expressed genes between the control and experimental conditions during the in the nest, eggs, early, mid and late stages (1674 genes). Red indicates positive values whereas blue indicates negative values.

That being said, while hierarchical clustering or PCA is useful for testing the carryover hypothesis, they are not as useful for testing the unique genes or additive hypotheses. Moreover, we would like to argue that our original analyses do, indeed, statistically test our hypotheses in rigorous ways, and we did not simply cherry pick genes. Let me explain:

1) In order to test the unique genes hypothesis, we took a statistical approach that allowed us to be very confident that a gene that we were calling “unique” was, indeed, really unique to a particular stage, and was not shared with other stages. Our primary concern was that a gene might appear to be unique to one stage, but it might not actually be unique to that stage because it just barely passed the cutoff for statistical significance in another stage. Therefore we wanted the statistical burden for inferring *uniqueness* to be quite high.

For example, for each particular stage (e.g. nest), we wanted to know what are the genes that are differentially expressed in the nest stage, but are really *not* differentially expressed the eggs, early, middle, late stages. We were worried that we

might mistakenly think that a gene was unique to the nest stage but it just barely passed the statistical threshold in the egg stage, for example. Therefore in order for a gene to make it onto the list of differentially expressed genes that were “unique” to the nest stage, we required that the gene had to be differentially expressed between the control and nest condition at $FDR < 0.01$ AND it had to be differentially expressed in all the other stages at $FDR > 0.2$. In other words, genes that were unique to the nest stage were differentially expressed in the nest stage but were definitely NOT differentially expressed in the eggs, early, middle or late stages. We added Figure 2 below as Supplementary Figure S1 to further explain the reasoning and method for this analysis.

Figure 2. Procedure for identifying genes that are “unique” to a particular set of genes (i.e. nonshared). a) The table shows hypothetical FDR values of the differential expression between the control and experimental condition in two different comparisons for 20 different genes. In order for a gene to be considered “unique” to a particular set, it had to be DE in that set at $FDR < 0.01$ AND it had to be clearly NOT differentially expressed in another set, i.e. at $FDR > 0.2$. In contrast, genes that were “shared” between sets were DE at $FDR < 0.01$ in both sets. b) Venn diagrams to illustrate how increasing the FDR threshold in the non-focal set reduces the size of the gene set that is unique to the focal set. These hypothetical venn diagrams show the number of DEGs in two different sets (1 and 2) and the overlap between them. In the top panel, the hashed area indicates genes that are DE in set 1 at $FDR < 0.01$ but are not DE in set 2 at $FDR < 0.01$. To guard against the possibility that some of the genes in the hashed area just barely passed the threshold for differential expression in set 2, we expanded the gene set in set 2 to include genes that were DE at $FDR < 0.2$, as indicated by the orange arrow. The hashed area now indicates genes that we consider to be unique to set 1; note that the size of the hashed area in the bottom panel is considerably smaller than the hashed area in the top panel, because the “unique” genes had to pass a more stringent filter for inclusion.

It is important to note that we did not lower the threshold in the focal stage to which the gene was unique, i.e. genes that were unique to the nest stage were, indeed, statistically differentially expressed between the control and nest condition at $FDR < 0.01$.

Instead, we changed the threshold in the *other*, nonfocal stages. We acknowledge that it is a bit unconventional to worry about this problem (but see Stein et al 2018), but we expect that as the community gets interested in nonshared genes (as opposed to just shared genes), there will be growing appreciation that the burden for showing that a gene is “unique” is different from the burden for showing that a gene is shared, i.e. that convincing evidence for nondiscovery might be different from convincing evidence for discovery.

2) In order to test the carryover hypothesis, we first tested for differential expression between the experimental and control condition at each stage using a stringent statistical cutoff (FDR < 0.01). We then used hypergeometric tests to statistically assess whether the number of genes that were shared among stages was greater than expected due to chance, considering the total number of genes in the background universe. We infer that significant (FDR < 0.05) overlap according to the hypergeometric test is evidence in support of the carryover hypothesis. This approach allowed us to test at a genome-wide level if the number of shared genes is greater than we would expect due to chance.

3) In order to test the additivity hypothesis, we used another approach. In order to qualify as an “added shared gene”, the gene had to pass several statistical thresholds, i.e. it had to 1) be differentially expressed during the stage of interest; 2) not be differentially expressed during any of the preceding stages; 3) be differently expressed in a subsequent stage. We then go on to show that according to two different analyses (ROAST and a simple Chi-Square test), the number of added shared genes showing a concordant expression pattern (i.e. consistently up- or down-regulated across stages) is much greater than we would expect due to chance genome-wide. We want to emphasize that we did not just find genes that fit the pattern we were looking for, instead we used a systematic procedure to identify those genes and then tested whether those genes exhibited a pattern that was statistically different from the null expectation. Note also that a concordant expression pattern was not part of the criterion used to identify the genes; it was statistically tested in an unbiased manner via two different methods.

Therefore we agree that given the richness of the dataset we probably could have picked a gene to illustrate any pattern we wanted to, but that’s not what we did. Instead, we did rigorous statistical testing to show that there are more genes that fit a pattern than we would expect due to chance (in the case of the carryover and additivity hypothesis), and/or that the genes really, really illustrate a particular pattern with high statistical confidence (in the case of the unique hypothesis). The MS now includes more text to clarify these points.

Minor comments.

Line 273. “however we used a more stringent FDR cutoff of FDR < 0.25 here instead of FDR < 0.20”. Two questions. 1 – Why were the different cutoffs used for the same type of analysis? For consistency, the same one should be used in both.

Good point. We now use the same cut-off in both analyses and have updated Table S7 accordingly; the main result that the functional terms are nonoverlapping between territorial challenge and paternal care is unchanged.

2 – I think $FDR < 0.25$ is actually less stringent than $FDR < 0.2$, unless I misunderstand the nature of FDR corrections.

We see how this is confusing and we now provide a supplementary figure (Supplementary Figure S1) to illustrate the procedure we used to find unique genes. $FDR < 0.25$ is actually more stringent than $FDR < 0.20$ in this approach because it allows more genes to be called as differentially expressed in the nonfocal set, i.e. by changing the FDR from 0.2 to 0.25 in the nonfocal condition, it reduces the number of genes that are considered to be “unique” to the focal condition.

Line 443. “We included genes with at least one counts per million (cpm) in at least two samples”. Two things. 1 – I think this should be “one count per million”.

Thanks for pointing this out; this is now corrected in the MS.

2 - Apologies for not noticing in the previous draft, but this is a somewhat problematic threshold as it does not correct for differences in gene length. Because it is based on strict read count, longer genes will have a higher likelihood of passing than shorter genes as they will simply have more mappable reads overall. The authors should use a method that corrects for gene length such as RPKM, TPM.

We normalized our gene counts using TMM normalization as recommended by edgeR before conducting tests for differential expression. We understand that different gene lengths can affect differential expression analysis, however gene length remains the same for all replicates, therefore it is expected to have little effect on differential expression analysis (Robinson et al 2010).

Line 490. Change to Orthogroups.

Corrected.

Reviewer #2 (Remarks to the Author):

I like this experiment more every time I read the manuscript.

There are a few previous Issues raised that I do not think were addressed adequately. These were not strong counter arguments, in what otherwise was a strong rebuttal:

Original - Figure 4. An overlap of 14 orthogroups produced an FDR of < 0.001 ? There were 2000 DEGs here and 3790 orthogroups. How many DEGs in the orthogroups for Ga and for Mm?

The test you describe is exact how I envisioned you did it. This just needs to be cleared up for me a bit more because it still seems unbelievable. From a list of 3790 elements, you sampled 904 elements (representing the Mm orthogroups that contain DEGs) and,

independently, 504 elements (representing the Ga orthogroups that contain DEGs). You then asked for the mean intersection of these two list from 10,000 independent draws? I do this 10 times and I get a mean of 117 +/- 9 elements. That suggests that 14 in the observed list is evidence of an underrepresentation of shared orthologs, rather than an enrichment. Is this what is being done?

No. Sorry for the confusion. We did not test for an overlap between differentially expressed genes between mouse (3570 genes within 924 orthogroups) and stickleback (2118 genes within 504 orthogroups) genes. Instead, we test for an overlap between mouse and stickleback **added shared genes** (356 stickleback genes within 90 orthogroups and 838 mouse genes within 265 orthogroups genes), with 14 orthogroups shared between them. In order to test whether those 14 shared orthogroups is greater than expected due to chance we employed a Monte Carlo based permutation approach. We did not use a regular hypergeometric test or regular permutation test here (at the orthogroup level) because each orthogroup contains more than one gene in both the stickleback and mouse genomes, and some of those genes were differentially expressed and others were not. Instead, we sampled the gene sets (356 and 838 genes) repeatedly (10^5) and with replacement from both species' universes and counted the overlaps at the orthogroup level. This overlap was then tested against the observed overlap to compute p-values. The results are in Figure 4. Note that the overlap never reaches 14 orthogroups. We modified the language slightly in the methods to clarify how the overlap was assessed.

Note in this MS we are only looking for commonalities between mouse and stickleback added shared genes and unique genes; the size of overlap between all mouse and stickleback DEGs is much larger. This is explained further in our response to Reviewer #3.

Figure 4. Overlap recovered while sampling gene sets randomly from both

stickleback and mouse universe 1e5 times. The y-axis shows the size of overlap; the x-axis shows the number of iterations.

Unless the “genomes” mentioned are not just restricted to gene represented in the orthogroups, but all genes for each organism.

The genomes mentioned were restricted to genes for each organism; we sampled repeatedly from mouse and stickleback gene universes. The universe consists of genes that were used to build linear models for each species.

Looking at the equation again, should you not be asking how many time t^* is less than, rather than greater than, t_i for a test of overrepresentation?

Sorry that was confusing but actually the equation does indicate that $t^* < t_i$ and the equation as written in the manuscript is the same as in Ernst 2004, page 682.

This seems to be a crux of the viability of Ga to be representative of vertebrates generally for parental care. I think this argument and its exact test needs to be fleshed out more for the reader.

Thank you for encouraging us to better explain the methods used in the orthogroup analysis, which we agree can be confusing. We clarified in the methods that n_1 and n_2 are gene set sizes.

Original - Line 458 - Why were such old read aligning programs used? Both TopHat2 and Bowtie2 have been superseded by newer, and supposedly much improved, programs by the same groups (Salzberg/Landmead), HiSAT2.

That you want to make this analysis comparable to an older paper is ok. That is, however, an argument for reanalysing the old data with the newer methods and not an argument for analysing new data with a dated method. I was thinking doing a simple comparison between mapping rates of the two programs and showing no real difference exist. Otherwise, there is no need to produce new methods that address known problems with older methods or keeping abreast of best practices within a field.

In response to this suggestion, we realigned reads for a subset of our samples using HiSAT2 and compared its alignment efficiency on features with TopHat2. In general, HiSAT2 aligned more reads than TopHat2. However, it only improved 3.5% on feature alignments compared to TopHat2. This is not altogether surprising considering that the primary advantage of HiSAT2 is computational efficiency and not in aligner performance. Since we've already paid for the computation to do TopHat, using a more efficient aligner doesn't really reap big rewards for this study. Importantly, feature counts from both of the tools are highly correlated. For example, Figure 5 shows the correlation between feature counts according to both tools for one sample ($r=0.99$). Since our paper

focuses on larger gene expression patterns instead of individual genes, and we get very similar alignment performances between the two versions, we do not think there is any real benefit to using the newer version.

Figure 5. Correlation between HiSAT2 (Y axis) and Tophat2 (X axis) feature counts.

Original - Line 471 – Why were the two brain parts analysed separately? Why not analyse everything together and use contrasts to pick out specific comparisons with the model containing ID as a random effect? limma, much the same group as edgeR, has the ability to do this and is used here for other things.

Wanting to test specific a priori hypotheses (aka, contrasts) within a model is not exclusive of running a full model. There is no need to talk about a 3 way interaction even if modelled, which I agree would be nigh impossible to interpret. Unless it is the only significant factor and even then it tells the reader that there are not strong main effects. Having all samples from the same individual in one model allows the variance to be partitioned better, certainly for the random effect of individual. That you did it this way before is a reason to reanalyse old data with a more complete model and not a reason to analyse new data with a less suitable model.

We have thought long and hard about this issue and indeed grappled with it when we first started analyzing the data from this experiment. We ultimately decided to treat the brain regions separately, and we stick by this decision, despite the important points raised by reviewer #2.

In our previous response to reviewers, we listed several of the reasons why we chose to analyze brain regions separately rather than in a combined model with individual as a random effect, e.g. 1) EdgeR assumes a negative binomial distribution (most appropriate

for these data) but does not fit random effects; 2) we did not have a priori hypotheses about how brain regions would differ between stages; 3) we wanted to compare with a previous study (although we acknowledge the reviewer's point that we could have reanalyzed the older study if we thought the method was better); 4) the study was sufficiently powerful that it did not need the extra power provided by a combined analysis; 5) we were not interested in the individual random effect per say, etc. So we will not belabor those again here.

But another really important point that we failed to mention previously is that brain region is by far the strongest signal in the dataset as a whole, and the dominant signal of brain region could distort our inferences about treatment and stage effects in a combined model. For example, there are over 10,000 genes that are differentially expressed at $FDR < 0.01$ between telencephalon and diencephalon in this dataset (compare this to the roughly hundreds of genes that differ in expression between stages within a brain region). The difference in gene counts across brain regions ranges from ~2000 to ~250,000 cpm, which is much greater variation than is observed between treatments or stages. This is entirely consistent with our understanding of the brain, namely that it is comprised of highly specialized nuclei, regions, subpopulations of cells, circuits, etc within the brain. We know that the same gene can serve very different functions in different brain regions and that the same gene can be expressed at very different levels in different parts of the brain.

This is relevant because such strong differences between brain regions are likely to strongly alter the estimates of dispersion in a combined model that includes brain region as a factor. This could cause inappropriate borrowing of dispersion between brain regions that could potentially distort the treatment and stage effects, which were our primary interest. By analyzing brain regions separately, we minimize distortion of the error term, and this method also takes care of the random effect of individual.

We also wish to acknowledge that a combined model is arguably more powerful than running two separate models for each brain region, but the sample sizes and coverage in this experiment are very good: our sequencing depth is 2-3 times more than usually recommended for the stickleback genome, therefore power is not an issue in our analysis.

Because our study was sufficiently powerful, and because we were not interested in determining how much variance could be attributed to the random effect of individual, we think that there simply doesn't appear to be much biological knowledge to be gained from fitting a combined model with a random effect. In other words, there are few "pros" to a combined model, but there are several "cons".

Original - Line 464 – Why were multi-mapped reads excluded?

No. One does not need to know where a read exactly comes from for DE analysis. Using "compatibility counts" for each read/transcript comparison is the entire basis of the next generation of pseudo-alignment DE programs (Salmon, Kallisto), which are performing at the level of or better than older DE programs. Not a major issue, just again something where this paper is not aligned with the leading edge of RNA-seq. I do not think it biases the analyses, just limits their power.

Thank you for this comment. As noted above, the sample size and sequencing depth in this experiment are very good therefore we chose to be conservative by not including multi-mapped reads.

Minor comments

Figure 1C is not mentioned in results in the “Figure 1” paragraph. I think it might worth mentioning that all the genes of Figure 1C had overall significance and that the stars for significance are not just a huge number of pairwise comparisons.

Thanks for pointing this out. We now refer to Figure 1c in the “figure 1” paragraph, and indicate what the stars represent in the main text as well as in the figure caption. Figure 1c is also discussed in the section “Neuromolecular pathways of parenting are not sex specific and are deeply conserved”.

Reviewer #3 (Remarks to the Author):

Overall, I think the analysis and presentation are improved. The authors satisfactorily responded to all of my comments in the first review.

Additional comments:

While the authors suggest that the changes are "analogous to the changes associated with pregnancy, parturition and the postpartum period in mammalian mothers.", the actual number of genes with overlap is small, even though the overlap is significant. However, the sets of genes with orthologs appears to be small based on STable 6 and the methods. Perhaps the authors can be more straightforward in the description of these results and lay out for the reader in the results the number of genes with orthologs that can actually be tested in that comparison.

Thank you for this suggestion. We acknowledge that the overlap might, at first, glance, appear to be small. However, we wish to emphasize that in this analysis we did not compare all of the DEGs associated with parental care between mouse and stickleback. Instead, we focused on a smaller list of genes that we were particularly interested in because their expression carried over across stages of care, i.e. the “added shared genes”. Instead of comparing all of the 3570 DEGs in mouse and the 2118 DEGs in stickleback, we compared the 838 added shared genes in mouse and the 356 added shared genes in stickleback. Therefore the resulting overlap between the two species is actually quite specific – these are genes associated with the enduring signal of care in both species. We have reworded sections of the MS to draw attention to this subtle but important point.

The reviewer might be interested to know that if we do compare the two species' DEG lists (not just the added shared genes), the results are also highly significant: The 3570 mouse DEGs are represented in 924 orthogroups and the 2118 stickleback DEGs are represented in 504 orthogroups. There are 159 orthogroups shared between them, and this overlap is significant at $P < 0.001$. We are presently working on a cross-species comparison of parental care at the neurogenomic level that will explore these patterns further; those results will be reported in another paper.

I also wonder what cut-off was used to consider a gene a gene an ortholog (p value, e value) and if it was too stringent. While all the information is in the manuscript, a casual reader may not realize that the number of genes with overlap are small and this may be because the number of orthologs called is small. This is especially important given that the observation regarding overlap with mouse/mammal maternal expression is highlighted in both the abstract and discussion of the manuscript, as an important discovery.

See response above – the number is small because this analysis focuses just on the added shared genes rather than all the DEGs.

One minor comment is regarding the following sentence: "even in the absence of parturition, postpartum ovulation and lactation and their associated hormone dynamics". Do the authors mean "postpartum ovulation"? I wonder if that sentence is meant to indicate postpartum events, other than ovulation, that proceed lactation?

Thanks for catching the error – this has been rephrased to “postpartum events”.

References cited

- L. R. Stein, S. A. Bukhari, A. M. Bell, Personal and transgenerational cues are nonadditive at the phenotypic and molecular level. *Nature Ecology & Evolution* **2**, 1306-1311 (2018).
- M. D. Ernst, Permutation Methods: A Basis for Exact Inference. *Statist. Sci.* **19**, 676-685 (2004).
- M. D. Robinson, D. J. McCarthy, G. K. Smyth, edgeR: a Bioconductor package for differential expression analysis of digital gene expression data. *Bioinformatics (Oxford, England)* **26**, 139-140 (2010).

Reviewers' comments:

Reviewer #1 (Remarks to the Author):

The authors have satisfied all my concerns, and I have no further suggestions. I would like to thank them for their careful response letter.

Reviewer #2 (Remarks to the Author):

I think the authors have adequately addressed most comments.

I have only one real comment: The molecular mechanisms between parental care of stickleback and mice are NOT deeply conserved. An overlap of 14 out of a possible 90 orthogroups might be statistically significant, but it not enough evidence to state "these results show that the molecular mechanisms of parental care are deeply conserved". That is a 15% overlap. According to the results of this paper, there are more non-overlapping orthogroups than overlapping orthogroups. Additionally, this is only true for a specific subset of genes associated with parental care in either species. There is not evidence provided for this being true of DEGs in general and it is explicitly not true of unique genes.

This is something that both Reviewer 3 and I mentioned. The response is that the overlap is statistically significant, which I now understand and agree with. That is, however, not a response to the underlying concern that the conclusions in the abstract and discussion headers, as stated, are more false than true. In response to Reviewer 3, the authors do report that the overlap of orthogroups containing DEG is 159/504 (32 %). This is more inline with the results needed to make the above statement, but it still means that here are more non-overlapping orthogroups than overlapping orthogroups.

There is enough evidence to say "that some of the molecular mechanisms of parental care are deeply conserved," but nothing more than that.

The manuscript should not be allowed forward without this concerns being addressed.

Permutation test. I see, the equation reports the proportion of iterations that falsify the hypothesis. In the results, I would also report the non-significant P value associated with the unique gene set for the sake of consistency.

Aligner. It is true that the single largest gain of HISAT2 was efficiency; however, there is also real gains in accuracy (see HISAT paper itself or most reviews of aligners, such as , Baruzzo et al. 2017 Nature Methods). You yourself report a 3.5% gain, which is larger than the ~1% I have seen with my own datasets when switching. That being said, I agree it is unlikely to really bias the results. As a pure sanity check for myself, I would make sure that there is not something fundamentally different about my DEG subset between HISAT and TopHat.

I would push back on two of your arguments. 1) Read depth is not a good way to increase power (Conesa et al. 2016 Genome Biology; Todd et al. 2016 Molecular Ecology) and 2) Five biological replicates is not enough for high statistical power. It is certainly sufficient, but not exceptional by any measure.

Default parameters, citations, and versions. If true, please put a sentence at the beginning of the bioinformatic section that all programs were used a default setting unless specified otherwise. HTSeq, Trimmomatic, and edgeR must be cited. FASTQC, HTSeq, Trimmomatic, and edgeR do not have version numbers.

Reviewer #3 (Remarks to the Author):

I think the authors did a good job responding to my comments, but I still don't see the numbers mentioned in their response in the actual manuscript files. I think they need to be explicit and state what they say in the response comments, so their readers understand. The number of genes compared are hard to figure out based on their presentation. I tried by looking at the tables and was not able to easily come up with those numbers.

They can say something like: We compared 838 added shared genes in mouse and 356 added shared genes in stickleback resulting in XX genes overlapping, which is significant ($P < XX$) based on permutation statistical test. The same type of statement should be done for the unique, even though they don't find significant changes.

-While overall the criteria used to define unique and added genes is much more clear, I still think this concept of the shared genes and unique genes is more methodological than based on a clear biological observation, and the paper would have been strengthened and be more clear if they had added a global analysis of the gene sets identified in the comparative studies.

-Why don't the authors provide a statistical cut-off for their orthoDB analyses?

Additional comments:

1) I encourage the authors to consider their statement on page 5, line 85-86. I doubt this study serves as a model for some of the life events they mention, even if they have behaviorally defined stages. For example, courtship steps and prey detection occur on a time scale of seconds-minutes. The progression through the steps are not likely to be events that are regulated at the gene expression level.

2) error in writing on line 190, page 10. I think it should be each of the added shared genes.

3) error in writing on line 225, page 12

4) The language regarding maternal and paternal care being "interchangeable" at the molecular level is an over interpretation. This is quite a stretch given the number of genes that overlap based on the analysis presented.

5) The sentences in the abstract that says fish and mammals parental care are "analogous" and that it is "deeply conserved" are too strong based on the results presented.

Dear Editor,

Thank you for the chance to revise our manuscript in response to the reviewers' comments. We thank the reviewers for their careful reading and attention to the manuscript; their suggestions have helped to improve the manuscript. Below our responses are in red text.

Best,
Alison

Reviewers' comments:

Reviewer #1 (Remarks to the Author):

The authors have satisfied all my concerns, and I have no further suggestions. I would like to thank them for their careful response letter.

Reviewer #2 (Remarks to the Author):

I think the authors have adequately addressed most comments.

I have only one real comment: The molecular mechanisms between parental care of stickleback and mice are NOT deeply conserved. An overlap of 14 out of a possible 90 orthogroups might be statistically significant, but it not enough evidence to state "these results show that the molecular mechanisms of parental care are deeply conserved". That is a 15% overlap. According to the results of this paper, there are more non-overlapping orthogroups than overlapping orthogroups. Additionally, this is only true for a specific subset of genes associated with parental care in either species. There is not evidence provided for this being true of DEGs in general and it is explicitly not true of unique genes.

This is something that both Reviewer 3 and I mentioned. The response is that the overlap is statistically significant, which I now understand and agree with. That is, however, not a response to the underlying concern that the conclusions in the abstract and discussion headers, as stated, are more false than true. In response to Reviewer 3, the authors do report that the overlap of orthogroups containing DEG is 159/504 (32 %). This is more inline with the results needed to make the above statement, but it still means that here are more non-overlapping orthogroups than overlapping orthogroups. There is enough evidence to say "that some of the molecular mechanisms of parental care are deeply conserved," but nothing more than that.

The manuscript should not be allowed forward without this concerns being addressed.

We agree; the interpretation is now rephrased as you suggest with changes on lines 31, 35, 221, 285, 401.

Permutation test. I see, the equation reports the proportion of iterations that falsify the hypothesis. In the results, I would also report the non-significant P value associated with the unique gene set for the sake of consistency.

Corrected on lines 267-270, line 558.

Aligner. It is true that the single largest gain of HISAT2 was efficiency; however, there is also real gains in accuracy (see HISAT paper itself or most reviews of aligners, such as , Baruzzo et al. 2017 Nature Methods). You yourself report a 3.5% gain, which is larger than the ~1% I have seen with my own datasets when switching. That being said, I agree it is unlikely to really bias the results. As a pure sanity check for myself, I would make sure that there is not something fundamentally different about my DEG subset between HISAT and TopHat.

Thank you for this suggestion. We did compare the results of the two analyses and they are largely in agreement with each other.

I would push back on two of your arguments. 1) Read depth is not a good way to increase power (Conesa et al. 2016 Genome Biology; Todd et al. 2016 Molecular Ecology) and 2) Five biological replicates is not enough for high statistical power. It is certainly sufficient, but not exceptional by any measure.

Point well taken.

Default parameters, citations, and versions. If true, please put a sentence at the beginning of the bioinformatic section that all programs were used a default setting unless specified otherwise.

Corrected line 473-4.

HTSeq, Trimmomatic, and edgeR must be cited.

Corrected on line 477, 481; FYI Trimmomatic was removed from the text because we did not trim the reads.

FASTQC, HTSeq, Trimmomatic, and edgeR do not have version numbers.

Corrected, see lines 468, 471, 477, 481, 486.

Reviewer #3 (Remarks to the Author):

I think the authors did a good job responding to my comments, but I still don't see the numbers mentioned in their response in the actual manuscript files. I think they need to be explicit and state what they say in the response comments, so their readers understand. The number of genes compared are hard to figure out based on their presentation. I tried by looking at the tables and was not able to easily come up with those numbers.

They can say something like: We compared 838 added shared genes in mouse and 356 added shared genes in stickleback resulting in XX genes overlapping, which is significant ($P < XX$) based on permutation statistical test. The same type of statement should be done for the unique, even though they don't find significant changes.

Thank you for pointing this out. We now include more of this information in the results section on lines 252-264.

-While overall the criteria used to define unique and added genes is much more clear, I still think this concept of the shared genes and unique genes is more methodological than based on a clear biological observation, and the paper would have been strengthened and be more clear if they had added a global analysis of the gene sets identified in the comparative studies.

Point well taken. We are in the process of completing a more comprehensive comparative analysis of the neurogenomics of parental care.

-Why don't the authors provide a statistical cut-off for their orthoDB analyses?

We assume that the reviewer is asking for the statistical cutoff for the overlap between the orthogroups, which is $FDR < 0.05$. This has been added to the text on line 558.

Additional comments:

1) I encourage the authors to consider their statement on page 5, line 85-86. I doubt this study serves as a model for some of the life events they mention, even if they have behaviorally defined stages. For example, courtship steps and prey detection occur on a time scale of seconds-minutes. The progression through the steps are not likely to be events that are regulated at the gene expression level.

That's a really good point - thanks. We removed text about behavioral changes that occur on short minutes-hours time scales (stages of courtship, stages of predator detection) and replaced them with slower changes that occur over the course of days/weeks/months, e.g. stages of dispersal, stages of pair-bonding; see lines 86-87.

2) error in writing on line 190, page 10. I think it should be each of the added shared genes.

Good catch! Corrected on line 193.

3) error in writing on line 225, page 12

Corrected on line 229– thanks.

4) The language regarding maternal and paternal care being "interchangeable" at the molecular level is an over interpretation. This is quite a stretch given the number of genes that overlap based on the analysis presented.

Replaced "interchangeable" with "share similarities". We made changes to the text in response to these comments on lines 31, 35, 221, 285, 401.

5) The sentences in the abstract that says fish and mammals parental care are "analogous" and that it is "deeply conserved" are too strong based on the results presented.

Rephrased, see also response to reviewer #2. We made changes to the text in response to these comments on lines 31, 35, 221, 285, 401.